# DEEP WATERMARKS
# FOR ATTRIBUTING GENERATIVE MODELS

## ABSTRACT

Generative models have enabled the creation of contents that are indistinguishable from those taken from the Nature. Open-source development of such models raised concerns about the risks in their misuse for malicious purposes. One potential risk mitigation strategy is to attribute generative models via watermarking. Current watermarking methods exhibit significant tradeoff between robust attribution accuracy and generation quality, and also lack principles for designing watermarks to improve this tradeoff. This paper investigates the use of latent semantic dimensions as watermarks, from where we can analyze the effects of design variables, including the choice of watermarking dimensions, watermarking strength, and the capacity of watermarks, on the accuracy-quality tradeoff. Compared with previous SOTA, our method requires minimum compute and is more applicable to large-scale models. We use StyleGAN2 and the latent diffusion model to demonstrate the efficacy of our method.

## 1 INTRODUCTION

Generative models can now create synthetic contents such as images and audios that are indistinguishable from those taken from the Nature (Karras et al., 2020; Rombach et al., 2022; Ramesh et al., 2022; Hawthorne et al., 2022). This pose serious threat when used as malicious attempt, such as disinformation (Breland, 2019) and malicious impersonation (Satter, 2019). Such potential threats delays the industrialization process of the generative model, as conservative model inventors hesitate to release their source code (Yu et al., 2020). For example in 2020, OpenAI refused to release the source code of their GPT-2 (Radford et al., 2019) model due to concerns over potential malicious attempts Brockman et al. (2020), additionally, the source codes of DALL-E (Ramesh et al., 2021) and DALL-E 2 (Ramesh et al., 2022) are also not released for the same reason Mishkin & Ahmad (2022).

One of the potential means of solution is model attribution (Yu et al., 2018; Kim et al., 2020; Yu et al., 2020), where a model distributor tweaks each user-end model so that they generate contents with model-specific watermarks. In practice, we consider the scenario where the model distributor or regulator maintain a database of user specific keys which corresponds to each users' downloaded model. When malicious attempts has been made, the regulator will be able to identify the user that's responsible for such attempts by attribution. Additionally, we assume the distributed model is white-box, which potentially makes a separate watermarking module appended on top of the generator trivial, as malicious user can simply remove such module from the network. Instead, we propose deep watermarking method that is free from this limitation by embedding the watermarking module directly into the generative model itself.

Formally, let a set of $n$ generative models be $\mathcal{G} := \{g_i(\cdot)\}_{i=1}^n$ where $g_i(\cdot) : \mathbb{R}^{d_z} \to \mathbb{R}^{d_x}$ is a mapping from an easy-to-sample distribution $p_z$ to a watermarked data distribution $p_{x,i}$ in the content space, and is parameterized by a binary-coded key $\phi_i \in \Phi := \{0,1\}^{d_\phi}$. Let $f(\cdot) : \mathbb{R}^{d_x} \to \Phi$ be a mapping that attributes contents to their source models. We consider four performance metrics of a watermarking mechanism: The *attribution accuracy* of $g_i$ is defined as

$$A(g_i) = \mathbb{E}_{z \sim p_z} \left[ \mathbb{1}\left(f(g_i(z)) == \phi_i\right)\right]. \tag{1}$$

The *generation quality* of $g_i$ measures the difference between $p_{x,i}$ and the data distribution used for learning $\mathcal{G}$, e.g., the Fréchet Inception Distance (FID) score (Heusel et al., 2017) for images.

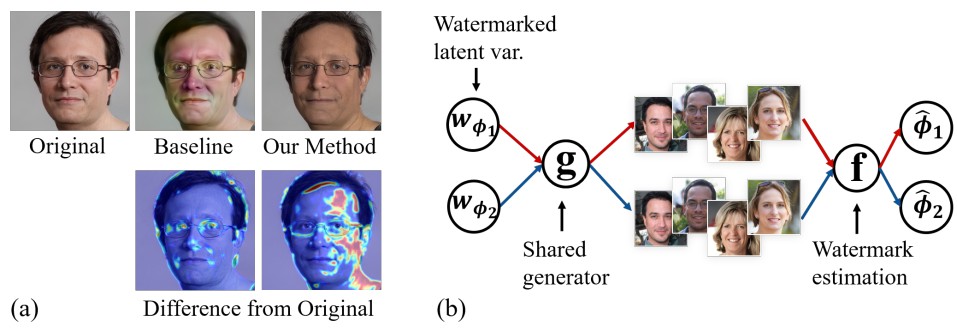

Figure 1: (a) Visual comparison between deep watermarking (our method) and shallow watermarking (Kim et al., 2020). Our method uses subtle semantic changes, rather than strong noises, to maintain attribution accuracy against image postprocesses. (b) Schematic of deep watermarking: The same generator $g$ and watermark estimator $f$ are used for all watermarked models. Our method thus requires minimal compute and is scalable to large latent diffusion models.

Inception score (IS) Salimans et al. (2016) is also measured for $p_{x,i}$ as additional *generation quality metrics*. *Watermark secrecy* is measured by the mean peak signal-to-noise ratio (PSNR) of individual images drawn from $p_{x,i}$. Compared with generation quality, this metric focuses on how obvious watermarks are rather than how well two content distributions match. Lastly, the *watermark capacity* is $n = 2^{d_\phi}$.

Existing watermarking methods exhibit significant tradeoff between attribution accuracy and generation quality (and watermark secrecy), particularly when countermeasures against dewatermarking attempts, e.g., image postprocesses, are taken into consideration. For example, Kim et al. (2020) use shallow watermarks for image generators in the form of $g_i(z) = g_0(z) + \phi_i$ where $g_0(\cdot)$ is an unwatermarked model, and show that $\phi_i$s have to significantly alter the original contents to achieve good attribution accuracy against image blurring, causing unfavorable drop in generation quality and watermark secrecy (Fig. 1(a)).

To improve this tradeoff, we investigate in this paper deep watermarks in the form of $g_i(\psi(z) + \phi_i) - g_0(\psi(z))$, where $w := \psi(z) \in \mathbb{R}^{d_w}$ contains disentangled semantic dimensions that allow a smoother mapping to the content space (Fig. 1). Such $\psi$ has been incorporated in popular models such as StyleGAN (SG) (Karras et al., 2019; 2020), where $w$ is the style vector, and latent diffusion models(LDM) (Rombach et al., 2022), where $w$ comes from a diffusion process. Existing studies on semantic editing showed that $\mathbb{R}^{d_w}$ consists of linear semantic dimensions (Härkönen et al., 2020; Zhu et al., 2021). Inspired by this, we hypothesize that using subtle yet semantic changes as watermarks will improve the robustness of attribution accuracy against image postprocesses, and thus investigate the performance of deep watermarks that are generated by perturbations along latent dimensions of $\mathbb{R}^{d_w}$. Specifically, we consider latent dimensions as eigenvectors of the covariance matrix of the latent distribution $p_w$, denoted by $\Sigma_w$.

**Contributions.** (1) We propose a novel intrinsic watermarking strategy that directly embed the watermarking module into the generative model, as a mean to achieve responsible white-box model distribution (2) We prove and empirically verify that there exists an intrinsic tradeoff between attribution accuracy and generation quality. This tradeoff is affected by watermark variables including the choice of the watermarking space, the watermark strength, and its capacity. Parametric studies on these variables for StyleGAN2 (SG2) and a Latent Diffusion Model(LDM) lead to improved accuracy-quality tradeoff from previous SOTA. In addition, our method requires negligible compute compared with previous SOTA, rendering it more applicable to popular large-scale models, including latent diffusion ones. (3) We show that using a postprocess-specific LPIPS metric for model attribution leads to further improved attribution accuracy against image postprocesses.

## 2 RELATED WORK

**Model attribution through watermark encoding and decoding.** Yu et al. (2020) propose to encode binary-coded keys into images through $g_i(z) = g_0([z, \phi_i])$ and to decode them via another learnable function. This requires joint training of the encoder and decoder over $\mathbb{R}^{d_z} \times \Phi$ to empirically balance attribution accuracy and generation quality. Since watermark capacity is usually

high (i.e., $2^{d_\phi}$), training is made tractable by sampling only a small subset of watermarks. Thus this method is computationally expensive and lacks a principled understanding of how the watermarking mechanism affects the accuracy-quality tradeoff. In contrast, our method does not require any additional training and mainly relies on simple principle component analysis of the latent distribution.

**Certifiable model attribution through shallow watermarks.** Kim et al. (2020) propose shallow watermarks $g_i(z) = g_0(z) + \phi_i$ and linear classifiers for attribution. These simplifications allow the derivation of sufficient conditions of $\Phi$ to achieve certifiable attribution of $\mathcal{G}$. Since the watermarks are added as noises rather than semantic changes coherent with the generated contents, high watermark strength becomes necessary to maintain attribution accuracy under postprocesses. While this paper does not provide attribution certification for deep watermarks, we discuss technical feasibility and challenges in achieving this goal.

**StyleGAN and low-rank subspace.** Our study focuses on popular image generation models which share an architecture rooted from SG: A uniform distribution is first transformed into a latent distribution ($p_w$), samples from which are then decoded into images. Härkönen et al. (2020) apply principal component analysis on $p_w$ distribution and found semantically meaningful editing directions. Zhu et al. (2021) use local Jacobian ($\nabla_w g$) to derive perturbations that enable local semantic editing of generated images, and show that such semantic dimensions are shared across the latent space. In this study, we show that the mean of the Gram matrix for local editing ($\mathbb{E}_{w \sim p_w}[\nabla_w g^T \nabla_w g]$) and the covariance of $w$ ($\Sigma_w$) are qualitatively similar in that both reveal major to minor semantic dimensions through their eigenvectors.

**GAN inversion.** The model attribution problem can be formulated as GAN inversion problem. A learning based inversion (Perarnau et al., 2016; Bau et al., 2019) optimizes parameters of encoder network which map a image to latent code $z$. On the other hand, optimization based inversion (Abdal et al., 2019; Huh et al., 2020) solve for latent code $z$ that minimizes distance metric between a given image and generated image $g(z)$. The learning based method is computationally more efficient in the inference stage comparing to optimization based method. However, optimization based GAN inversion achieves superior quality of latent interpretation, which can be referred to as the quality-time trade-off (Xia et al., 2022). In our method, we utilized the optimization based inversion, as faithful latent interpretation is critical in our application. To further enforce faithful latent interpretation, we incorporate existing techniques, e.g., parallel search, to solve this non-convex problem, but uniquely exploit the fact that watermarks are small latent perturbations to enable analysis on the accuracy-quality tradeoff.

## 3 METHODS

### 3.1 NOTATIONS AND PRELIMINARIES

**Notations.** For $x \in \mathbb{R}^n$ and $A \in \mathbb{R}^{n \times m}$, denote by $proj_A x$ the projection of $x$ to $span(A)$, and by $A^\dagger$ the pseudo inverse of $A$. For parameter $a$, we denote by $\hat{a}$ its estimate and $\epsilon_a = \hat{a} - a$ the error. $\nabla_x f$ is the gradient of $f$ with respect to $x$, $\mathbb{E}_{x \sim p_x}[\cdot]$ is an expectation over $p_x$, and $tr(B)$ (resp. $det(B)$) is the trace (resp. determinant) of $B \in \mathbb{R}^{n \times n}$. $diag(\lambda) \in \mathbb{R}^{n \times n}$ diagonalizes $\lambda \in \mathbb{R}^n$.

**Deep watermarks.** All contemporary generative models, e.g., SG2 and LDM, consist of a disentanglement mapping $\psi : \mathcal{R}^{d_z} \to \mathcal{R}^{d_w}$ from an easy-to-sample distribution $p_z$ to a latent distribution $p_w$, followed by a generator $g : \mathcal{R}^{d_w} \to \mathcal{R}^{d_x}$ that maps $w$ to the content space. In particular, $\psi$ is a multilayer perception network in SG, and a diffusion process in a diffusion model. Existing studies showed that linear perturbations along principal components of $\nabla_w g$ enables semantic editing, and such perturbation directions are often applicable over $w \sim p_w$ (Härkönen et al., 2020; Zhu et al., 2021). Indeed, instead of local analysis on the Jacobian, Härkönen et al. (2020) showed that principal component analysis directly on $p_w$ also reveals semantic dimensions. This paper follows these existing findings and uses a subset of semantic dimensions as watermarks. Specifically, let $U \in \mathbb{R}^{d_w \times (d_w - d_\phi)}$ and $V \in \mathbb{R}^{d_w \times d_\phi}$ be orthonormal and complementary. Given random seed $z \in \mathbb{R}^{d_z}$, user-specific key $\phi \in \mathbb{R}^{d_\phi}$, and strength $\sigma \in \mathbb{R}$, let $\alpha = U^\dagger proj_U \psi(z) \in \mathbb{R}^{d_w - d_\phi}$, the watermarked latent variable is

$$w_\phi(\alpha) = U\alpha + \sigma V\phi, \tag{2}$$

where $\alpha \sim p_\alpha$ and $p_\alpha$ is induced by $p_w$. Then, user can generate watermarked images, $g(w_\phi(\alpha))$. The choice of $(U, V)$ and $\sigma$ affects the attribution accuracy and generalization performance, which we analyze in Sec. 3.2.

**Attribution.** To decode user-specific key from the image $g(w_\phi(\alpha))$, we formulate an optimization problem:

$$\min_{\hat{\alpha}, \hat{\phi}} \quad l\left(g(w_{\hat{\phi}}(\hat{\alpha})), g(w_\phi(\alpha))\right)$$

$$s.t. \quad \hat{\alpha}_i \in [\alpha_{l,i}, \alpha_{u,i}], \quad \forall i = 1, ..., d_w - d_\phi.$$

Through experiments, we discovered that attribution accuracy can be improved by constraining $\alpha$. Here the upper and lower bounds of $\alpha$ are chosen based on the empirical limits observed from $p_\alpha$. While $l_2$ norm is used for analysis in Sec.3.2, here we minimize LPIPS (Zhang et al., 2018) which measures the perceptual difference between two images. In practice, we introduce a penalty on $\hat{\alpha}$ with large enough Lagrange multipliers and solve the resulting unconstrained problem. To avoid convergence to unfavorable local solutions, we also employ parallel search with n initial guesses of $\hat{\alpha}$ drawn through Latin hypercube sampling (LHS).

## 3.2 Accuracy-Quality Tradeoff

**Attribution accuracy.** Define $J_w = \nabla g(w)$, $H_w = J_w^T J_w$. Let $\bar{H}_w = \mathbb{E}_{w \sim p_w}[H_w]$ be the mean Gram matrix, and $\bar{H}_\phi = \mathbb{E}_{\alpha \sim p_\alpha}[H_{w_\phi(\alpha)}]$ be its watermarked version. Let $l : \mathbb{R}^{d_x} \times \mathbb{R}^{d_x} \to \mathbb{R}$ be a distance metric between two images, $(\hat{\alpha}, \hat{\phi})$ the estimates. To analyze how $(V, U)$ affects the attribution accuracy, we use the following simplifications and assumptions: **(A1)** $l(\cdot, \cdot)$ is the $l_2$ norm. **(A2)** Both $||\epsilon_\alpha||$ and $\sigma$ are small. In practice we achieve small $||\epsilon_\alpha||$ through parallel search (see Appendix B). **(A3)** Since our focus is on $\epsilon_\phi$, we further assume that the estimation of $\alpha$, denoted by $\hat{\alpha}(\alpha)$, is independent from $\phi$, and $\epsilon_\alpha$ is constant. This allows us to ignore the subroutine for computing $\hat{\alpha}(\alpha)$ and turns the estimation problem to an optimization with respect to only $\epsilon_\phi$. Formally, we have the following proposition (see Appendix A.1 for proof):

**Proposition 1.** $\exists\, c > 0$ *such that if* $\sigma \le c$ *and* $||\epsilon_\alpha||_2 \le c$*, the watermark estimation problem*

$$\min_{\hat{\phi}} \quad \mathbb{E}_{\alpha \sim p_\alpha}\left[\|g(w_{\hat{\phi}}(\hat{\alpha}(\alpha))) - g(w_\phi(\alpha))\|_2^2\right]$$

*has an error* $\epsilon_\phi = -(\sigma^2 V^T \bar{H}_\phi V)^{-1} V^T \bar{H}_\phi U \epsilon_\alpha$.

**Remarks:** (1) Similar to classic design of experiment, one can reduce $||\epsilon_\phi||$ by maximizing $det(V^T \bar{H}_\phi V)$, which sets columns of $V$ as the eigenvectors associated with the largest $d_\phi$ eigenvalues of $\bar{H}_\phi$. However, $\bar{H}_\phi$ is neither computable because $\phi$ is unknown during the estimation, nor is it tractable because $J_{w_\phi(\alpha)}$ is large in practice. To this end, we propose to use the covariance of $p_w$, denoted by $\Sigma_w$, to replace $\bar{H}_\phi$ in experiments. In Appendix C, we support this approximation empirically by showing that $\Sigma_w$ and $\bar{H}_w$ (the non-watermarked mean Gram matrix) are qualitatively similar in that the principal components of both matrices offer disentangled semantic dimensions. (2) Let the $k$th largest eigenvalue of $\bar{H}_w$ be $\gamma_k$. By setting columns of $V$ as the eigenvectors of $\bar{H}_w$ associated with the largest $d_\phi$ eigenvalues, and by noting that $\hat{\phi}$ is accurate only when all of its elements match with $\phi$ (equation 1), the worst-case estimation error is governed by $\gamma_{d_\phi}^{-1}$. This means that higher key capacity, i.e., larger $d_\phi$, leads to worse attribution accuracy. (3) From the proposition, $\epsilon_\phi = 0$ if $V$ and $U$ are complementary sets of eigenvectors of $\bar{H}_\phi$. In practice this decoupling between $\epsilon_\phi$ and $\epsilon_\alpha$ cannot be achieved due to the assumptions and approximations we made.

**Generation quality.** For analysis purpose, we approximate the original latent distribution $p_w$ by $w = \mu + U\alpha + V\beta$ where $\alpha \sim \mathcal{N}(0, diag(\lambda_U))$, $\beta \sim \mathcal{N}(0, diag(\lambda_V))$, and $\mu = \mathbb{E}_{w \sim p_w}[w]$. $\lambda_U \in \mathbb{R}^{d_w - d_\phi}$ and $\lambda_V \in \mathbb{R}^{d_\phi}$ are calibrated to match $p_w$. Denote $\lambda_{V,max} = \max_i\{\lambda_{V,i}\}$. A latent distribution watermarked by $\phi$ is similarly approximated as $w_\phi = \mu + U\alpha + \sigma V\phi$. With mild abuse of notation, let $g$ be the mapping from the latent space to a feature space (usually defined by an Inception network in FID) and is continuously differentiable. Let the mean and covariance matrix of $w_i$ be $\mu_i$ and $\Sigma_i$, respectively. Denote by $\bar{H}_U = \mathbb{E}_\alpha[J_{\mu+U\alpha}^T J_{\mu+U\alpha}]$ the mean Gram matrix in the subspace of $U$, and let $\gamma_{U,max}$ be the largest eigenvalue of $\bar{H}_U$. We have the following proposition to upper bound $||\mu_0 - \mu_1||_2^2$ and $|tr(\Sigma_0) - tr(\Sigma_1)|$, both of which are related to the FID score for measuring the generation quality (see Appendix A.2 for proof):

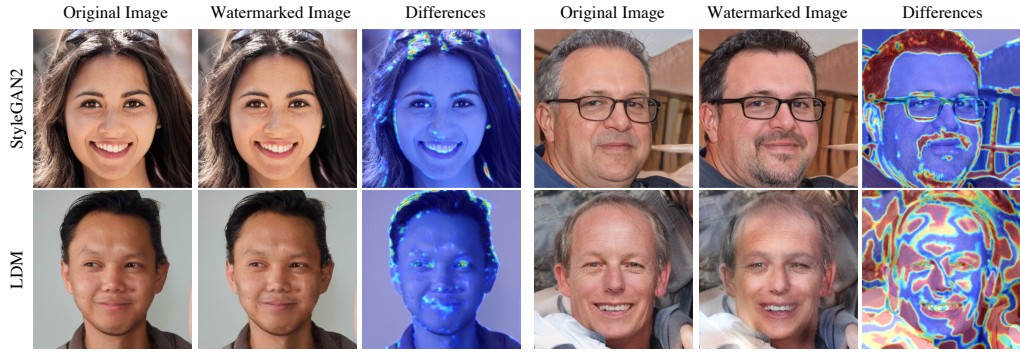

(a) Watermarking along Minor PCs  (b) Watermarking along Major PCs

Figure 2: Visualization of watermarks along minor and major principal components of the covariance of the latent distribution. (Top) StyleGAN2. (Bottom) Latent Diffusion Model (LDM).

**Proposition 2.** *For any $\tau > 0$ and $\eta \in (0, 1)$, $\exists\, c(\tau, \eta) > 0$ and $\nu > 0$, such that if $\sigma \leq c(\tau, \eta)$ and $\lambda_{V,i} \leq c(\tau, \eta)$ for all $i = 1, ..., d_\phi$, then $\|\mu_0 - \mu_1\|_2^2 \leq \sigma^2 \gamma_{U,max} d_\phi + \tau$ and $|tr(\Sigma_0 - \Sigma_1)| \leq \lambda_{V,max} \gamma_{U,max} d_\phi + 2\nu\sigma\sqrt{d_\phi} + \tau$ with probability at least $1 - \eta$.*

**Remarks:** Recall that for improving attribution accuracy, a practical approach is to choose $V$ as eigenvectors associated with the largest eigenvalues of $\Sigma_w$. Notices that with the approximated distribution with $\alpha \sim \mathcal{N}(0, diag(\lambda_U))$ and $\beta \sim \mathcal{N}(0, diag(\lambda_V))$, $\Sigma_w = diag([\lambda_U^T, \lambda_V^T]^T)$. On the other hand, from Proposition 2, generation quality improves if we minimize $\lambda_{V,max}$ by choosing $V$ according to the smallest eigenvalues of $\Sigma_w$. In addition, smaller key capacity ($d_\phi$) and lower strength ($\sigma$) also improve the generation quality. Propositions 1 and 2 together reveal the intrinsic accuracy-quality tradeoff.

**Watermark secrecy.** Lastly, the analysis on watermark secrecy is straight forward using the same proof techniques: We note that PSNR is a monotonically decreasing function of the MSE $\|g(\mu + U\alpha) - g(\mu + U\alpha + \sigma V\phi)\|_2^2$ and therefore we can use the following proposition to analyze the effect of watermark variables on secrecy (proof in Appendix A.3):

**Proposition 3.** *For any $\tau > 0$, $\exists\, c(\tau) > 0$ such that if $\sigma \leq c(\tau)$,*

$$\mathbb{E}_{\alpha \sim p_\alpha} \left[ \|g(\mu + U\alpha) - g(\mu + U\alpha + \sigma V\phi)\|_2^2 \right] \leq \sigma^2 \gamma_{U,max} d_\phi + \tau.$$

## 4 EXPERIMENTS

In this section, we present empirical evidence of the accuracy-quality tradeoff and show improved tradeoff from previous SOTA by using deep watermarks. Experiments are conducted for both with and without a combination of postprocesses including (image noising, blurring, and JPEG compression transformation, and their combination).

### 4.1 EXPERIMENT SETTINGS

**Models, data, and metrics.** We conduct experiments on SG2 (Karras et al., 2020) and LDM (Rombach et al., 2022) models trained on various datasets including FFHQ (Karras et al., 2019), AFHQ-Cat, and AFHQ-Dog (Choi et al., 2020). Generation quality is measured by the Frechet Inception distance (FID) (Heusel et al., 2017) and inception score (IS) (Salimans et al., 2016), attribution accuracy by equation 1, and watermark secrecy by PSNR.

**Deep watermark dimensions.** The dimensions of the semantic latent spaces of SG2 and LDM are 512 and 12,288, respectively. For both models we approximate $\Sigma_w$ using 10K samples drawn from $p_w$. We define watermark dimensions $V$ as a subset eigenvectors of $\Sigma_w$ associated with consecutive eigenvalues: $V := PC[i : j]$, where $PC$ is the full set of principal components of $\Sigma_w$ sorted by their variances in the descending order, and $i$ and $j$ represent the starting and ending indices of the subset.

**Attribution.** To compute the empirical accuracy, we use 1K samples drawn from $p_z$ for each watermark $\phi$, and use 1K watermarks where each bit is drawn independently from a Bernoulli distri-

bution with $p = 0.5$. In Table 1, we show that both constraints on $\hat{\alpha}$ and parallel search with 20 initial guesses improve the empirical attribution accuracy across models and datasets. Notably, constrained estimation is essential for successful attribution of LDMs. In these experiments, $V$ is chosen as the eigenvectors associated with the 64 smallest eigenvalues of $\Sigma_w$ as a worst-case scenario for attribution accuracy.

Table 1: Attribution accuracy and generation quality of the proposed method. FID-BL is the baseline FID score. $\uparrow$ ($\downarrow$) indicates higher (lower) is desired. Standard deviation is in parenthesis.

| Model | Dataset | Attribution Accuracy | | | Image Quality | | | |
|---|---|---|---|---|---|---|---|---|
| | | Ours | w/o $\alpha$-reg | w/o LHS | FID-BL | FID$\downarrow$ | IS $\uparrow$ | PSNR $\uparrow$ |
| SG2 | FFHQ | **0.983** | 0.877 | 0.711 | 7.24 | 8.59 | 4.96 | 32.5 (1.65) |
| SG2 | AFHQ Cat | **0.993** | 0.991 | 0.972 | 6.35 | 7.87 | 12.39 | 38.14 (1.79) |
| SG2 | AFHQ Dog | **0.999** | 0.998 | 0.981 | 3.80 | 5.36 | 12.39 | 37.7 (1.91) |
| LDM | FFHQ | **0.996** | 0.364 | 0.872 | 12.34 | 13.63 | 4.35 | 31.22 (1.90) |

## 4.2 Attribution performance without postprocessing

We present generation quality results in Table 1. Since the least variant principal components are used as watermarks, generation quality (FID) and watermark secrecy (PSNR) are preserved. We note that a PSNR value $\geq 30$ db is conventionally considered as acceptable for watermarks (Mahto & Singh, 2021). The results suggest that when image postprocesses are not considered as a potential threat model, the attribution accuracy, generation quality, capacity ($2^{64}$), and watermark secrecy are all acceptable using the proposed method. Fig. 2 visualizes and compares deep watermarks generated from small vs. large eigenvalues of $\Sigma_w$. Watermarks corresponding to small eigenvalues are non-semantic, while those to large eigenvalues create semantic changes. We will later show that semantic yet subtle (perceptually insignificant) watermarks are necessary to counter image postprocesses.

**Accuracy-quality tradeoff.** Table 2 summarizes the tradeoff when we vary the choice of $V$ and the strength $\sigma$, while fixing the watermark length $d_\phi$ to 64. Then in Table 3 we sweep $d_\phi$ while keeping $V$ as PCs associated with smallest eigenvalues of $\Sigma_w$ and $\sigma = 1$. The experiments are conducted on SG2 and LDM on the FFHQ dataset. The empirical results in Table 2 are consistent with our analysis: Accuracy decreases while generation quality improves when $V$ is moved from major to minor principal components. For watermark strength, however, we observe that the positive effect of strength on the accuracy, as predicted by Proposition 1, is only limited to small $\sigma$. This is because larger $\sigma$ causes pixel values to go out of bound, causing loss of information. In Table 3, we summarize the attribution accuracy, FID, and PSNR score under 32- to 128-bit keys. Accuracy and generation quality, in particular the latter, are both affected by $d_\phi$ as predicted.

## 4.3 Watermark performance with postprocessing

We now consider more realistic scenarios where generated images are postprocessed, either maliciously as an attempt to remove the watermarks or unintentionally, before they are attributed. Under this setting, our method achieves better accuracy-quality tradeoff than shallow watermarking under two realistic settings: (1) when noising an JPEG compression are used as *unknown* postprocesses, and (2) when *the set of postprocesses*, rather than the ones that are actually chosen, is known.

**Postprocesses.** To keep our solution realistic, we solve the attribution problem by assuming that the potential postprocesses are unknown:

$$\min_{\hat{\alpha},\hat{\phi}} \quad l\left(g(w_{\hat{\phi}}(\hat{\alpha})), T(g(w_\phi(\alpha)))\right)$$

$$s.t. \quad \hat{\alpha}_i \in [\alpha_{l,i}, \alpha_{u,i}], \quad \forall i = 1, ..., d_w - d_\phi.$$

where $T : R^{d_x} \to R^{d_x}$ is a postprocess function, and $T(g(w_\phi(\alpha)))$ is a given image from which the watermark is to be estimated. We assume that $T$ does not change the image in a semantically meaningful way, because otherwise the value of the image for either an attacker or a benign user will be lost. Since our method adds semantically meaningful perturbations to the images, we expect such

Table 2: Tradeoff between attribution accuracy (Att.) and generation quality (FID, IS) under different watermarking directions (PC) and strength ($\sigma$).

| StyleGAN2 | $\sigma = 0.6$ | | | $\sigma = 1.0$ | | | $\sigma = 6.0$ | | |
|---|---|---|---|---|---|---|---|---|---|
| | Att. ↑ | FID ↓ | IS ↑ | Att. ↑ | FID ↓ | IS ↑ | Att. ↑ | FID ↓ | IS ↑ |
| PC[0:64] | 0.99 | 129.0 | 1.23 | 0.99 | 110.8 | 1.59 | 0.99 | 101.3 | 4.31 |
| PC[128:192] | 0.98 | 8.5 | 4.93 | 0.99 | 8.7 | 4.92 | 0.99 | 39.2 | 3.94 |
| PC[256:320] | 0.98 | 8.6 | 4.96 | 0.99 | 9.1 | 4.87 | 0.96 | 31.1 | 3.90 |
| PC[448:512] | 0.97 | 8.1 | 4.99 | 0.98 | 8.5 | 4.96 | 0.90 | 26.3 | 4.75 |
| LDM | $\sigma = 1.0$ | | | $\sigma = 2.0$ | | | $\sigma = 3.0$ | | |
| | Att. ↑ | FID ↓ | IS ↑ | Att. ↑ | FID ↓ | IS ↑ | Att. ↑ | FID ↓ | IS ↑ |
| PC[0:64] | 0.99 | 33.62 | 3.84 | 0.99 | 33.17 | 3.70 | 0.99 | 34.07 | 3.82 |
| PC[1000:1064] | 0.77 | 13.32 | 4.37 | 0.99 | 13.75 | 4.40 | 0.99 | 16.03 | 4.35 |
| PC[2000:2064] | 0.32 | 13.17 | 4.45 | 0.99 | 13.63 | 4.35 | 0.99 | 15.74 | 4.34 |
| PC[3000:3064] | 0.12 | 12.98 | 4.43 | 0.97 | 13.61 | 4.45 | 0.99 | 15.44 | 4.48 |
| PC[4000:4064] | 0.00 | 12.77 | 4.35 | 0.96 | 13.61 | 4.42 | 0.99 | 15.41 | 4.41 |

Table 3: Attribution accuracy and generation quality for different watermark lengths. FID-BL is baseline FID score. Standard deviation is in parenthesis. ↑ (↓) indicates higher (lower) is desired. Standard deviation is in parenthesis.

| Key Length | Accuracy ↑ | FID-BL | FID ↓ | PSNR ↑ | IS ↑ |
|---|---|---|---|---|---|
| 32 | 0.982 | 7.24 | 8.49 | 36.3 (1.94) | 4.90 |
| 64 | 0.983 | 7.24 | 8.59 | 32.5 (1.65) | 4.96 |
| 96 | 0.981 | 7.24 | 9.50 | 29.2 (1.95) | 4.93 |
| 128 | 0.973 | 7.24 | 9.61 | 27.07 (1.80) | 4.91 |

deep watermarks to be more robust to postprocesses than shallow ones added directly to images, and will lead to improved attribution accuracy. To test this hypothesis, we consider four types of postprocesses: `Noising`, `Blurring`, `JPEG` and `Combo`. `Noising` adds a Gaussian white noise of standard deviation randomly sample from $U[0, 0.1]$. `Blurring` uses a randomly selected Gaussian kernel size from [3, 7, 9, 16, 25] and standard deviation of [0.5, 1.0, 1.5, 2.0]. We randomly sample the `JPEG` quality from [80, 70, 60, 50]. These parameters are chosen to be mild so that images do not lose their semantic contents. And `Combo` randomly chooses a subset of the three through a binomial distribution with $p = 0.5$ and uses the same postprocess parameters.

**Modified LPIPS metric.** In addition to testing the worst-case scenario where postprocesses are completely unknown, we also consider cases where they are known. While this is unrealistic for individual postprocesses, it is worth investigating when we assume that the set of postprocesses, rather than the ones that are actually chosen, is known. Within this scenario, we show that modifying LPIPS according to the postprocess improves the attribution accuracy. To explain, LPIPS is originally trained on a so-called "two alternative forced choice" (2AFC) dataset. Each data point of 2AFC contains three images: a reference, $p_0$, and $p_1$, where $p_0$ and $p_1$ are distorted in different ways based on the reference. A human evaluator then ranks $p_0$ and $p_1$ by their similarity to the reference. Here we propose the following modification to the dataset for training a postprocess-specific metrics: Similar to 2AFC, for each data point, we draw a reference image $x$ from the default generative model to be watermarked, and define $p_0$ as the watermarked version of $x$. $p_1$ is then a postprocessed version of $x$ given a specific $T$ (or random combinations for `Combo`). To match with the setting of 2AFC, we sample $64 \times 64$ patches from $x$, $p_0$, and $p_1$ as training samples. We then rank patches from $p_1$ as being more similar to those of $x$ than $p_0$. With this setting, the resulting LPIPS metric becomes more sensitive to watermarks than mild postprocesses. The detailed training of the modified LPIPS follows the vgg-lin configuration in Zhang et al. (2018). It should be noted that unlike previous SOTA where watermarks Kim et al. (2020) or encoder-decoder models Yu et al. (2020) are retrained based on the known attacks, our watermarking mechanism, and therefore generalization performance, are agnostic to postprocesses.

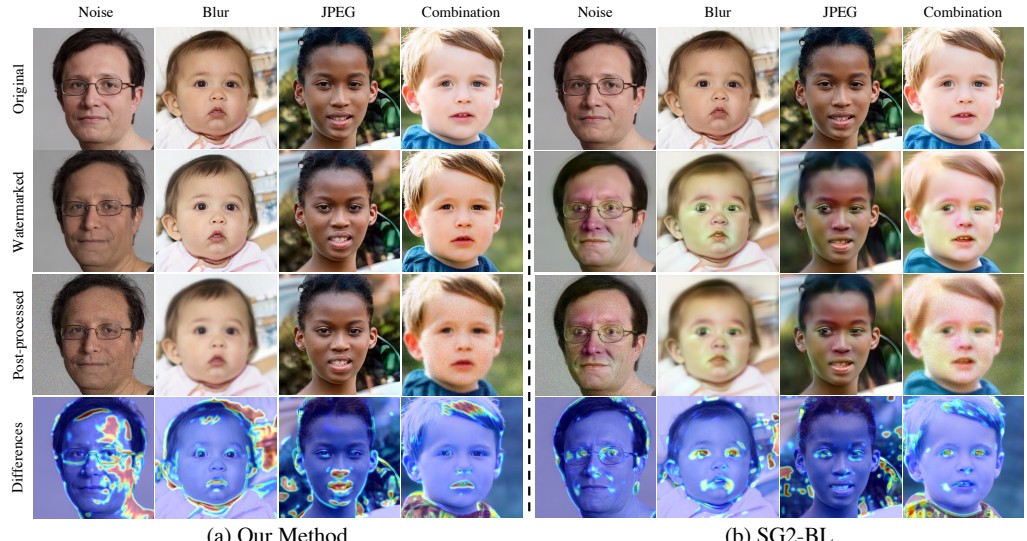

Figure 3: **Comparison on generation quality between our method and the baseline with similar attribution accuracy** The first row shows original images generated without watermarking. Each images in the second row represents robustly watermarked images against corresponding post-processes. The next row illustrates post-processed images. The last row depicts the differences between original (the first row) and watermarked images (the second row) using heatmap. Even if our method shows large pixel value changes, the watermarks are not perceptible comparing with baseline method (see second row).

**Accuracy-quality tradeoff.** We summarize watermark performance metrics on SG2 and FFHQ in Table 4. The attribution accuracy reported here are estimated using the strongest parameters of each attack. For `Combo`, we use sequentially apply `Blurring`+`Noising`+`JPEG` as a deterministic worst-case attack. To estimate attribution accuracy, we solved the estimation problem in equation 4.3 where postprocesses are applied. **The proposed method:** We choose $V$ as a subset of 32 consecutive eigenvectors of $\Sigma_w$ starting from the 1th, 17th, and 33th eigenvectors, denoted respectively by PC[0:32], PC[16:48], and PC[32:64] in the table. Watermarking strength $\sigma$ is set to 3. Attribution results from both a standard and a postprocess-specific LPIPS metric are reported in the UK (unknown) and KN (known) columns, respectively. Accuracies for our method are computed based on 100 random watermark samples from $2^{32}$, each with 100 random generations. **The baseline:** We compare with a shallow watermarking method from Kim et al. (2020) (denoted by BL). When the postprocesses are known, BL performs postprocess-specific computation to derive shallow watermarks that are optimally robust against the known postprocess. Results in UK and KN columns for BL are respectively without and with the postprocess-specific watermark computation. BL accuracies are computed based on 10 watermarks, each with 100 random generations.

It is worth noting that the shallow watermarking method is not as scalable as ours, and increasing the key capacity decreases the overall attribution accuracy (see (Kim et al., 2020)). Also recall that the key length affects attribution accuracy (Proposition 1). Therefore, we conduct a fairer comparison to highlight the advantage of our method. Here we choose a subset of watermarks $PC[32:40]$ (256 watermarks) and report performance in Table 5, where accuracies are computed using the same settings as before. Visual comparisons between our method ($PC[32:40]$) and the baseline can be found in Fig. 3: To maintain attribution accuracy, high strength shallow watermarks, in the form of color patches, are needed around eyes and noses, and significantly lower the generation quality. In comparison, our method uses semantic changes that are robust to postprocesses. The choice of the semantic dimensions, however, need to be carefully chosen for the watermark to be perceptually subtle.

**Watermark secrecy.** In all experiments, our method has worse watermark secrecy than the baseline according to PSNR. This is because PSNR measures pixel-wise differences, and thus does not favor semantic changes as in our method. Nonetheless, we argue that our method have better secrecy when compared with the baseline, because subtle semantic changes across images are harder to be recognized (and thus removed) than common artifacts brought by shallow watermarking (see Fig. 3).

Table 4: Comparison on accuracy-quality tradeoff between proposed and baseline methods under image postprocesses. StyleGAN2 and FFHQ. Watermarking strength $\sigma = 3$. The FID score of baseline method is 96.24. KN (UK) stands for when attributability is measured with (without) the knowledge of attack. Standard deviation is in parenthesis.

| Metric | Model | Blurring | | Noising | | JPEG | | Combo | |
|---|---|---|---|---|---|---|---|---|---|
| - | - | UK | KN | UK | KN | UK | KN | UK | KN |
| Att. ↑ | BL | 0.85 | 0.88 | 0.85 | 0.87 | 0.87 | 0.89 | 0.83 | 0.88 |
| | PC[0:32] | 0.79 | 0.99 | 0.99 | 0.99 | 0.98 | 0.99 | 0.82 | 0.99 |
| | PC[16:48] | 0.56 | 0.92 | 0.95 | 0.99 | 0.98 | 0.99 | 0.42 | 0.88 |
| | PC[32:64] | 0.32 | 0.83 | 0.93 | 0.98 | 0.98 | 0.99 | 0.26 | 0.79 |
| PSNR ↑ | BL | 21.36 (1.51) | | 21.57 (1.88) | | 21.44 (1.64) | | 22.49 (1.57) | |
| | PC[0:32] | 10.70 (0.56) | | 10.70 (0.56) | | 10.70 (0.56) | | 10.70 (0.56) | |
| | PC[16:48] | 13.56 (1.81) | | 13.56 (1.81) | | 13.56 (1.81) | | 13.56 (1.81) | |
| | PC[32:64] | 14.52 (1.31) | | 14.52 (1.31) | | 14.52 (1.31) | | 14.52 (1.31) | |
| IS ↑ | BL | 2.86 | | 3.02 | | 2.91 | | 2.90 | |
| | PC[0:32] | 2.93 | | 2.93 | | 2.93 | | 2.93 | |
| | PC[16:48] | 4.35 | | 4.35 | | 4.35 | | 4.35 | |
| | PC[32:64] | 4.50 | | 4.50 | | 4.50 | | 4.50 | |
| FID ↓ | BL | 99.05 | | 93.04 | | 97.70 | | 100.15 | |
| | PC[0:32] | 102.26 | | 102.26 | | 102.26 | | 102.26 | |
| | PC[16:48] | 31.25 | | 31.25 | | 31.25 | | 31.25 | |
| | PC[32:64] | 27.50 | | 27.50 | | 27.50 | | 27.50 | |

Table 5: Accuracy-quality tradeoff under Combo attack. $V$ defined as the 8, 16, and 32 eigenvectors of $\Sigma_w$ starting from the 33th eigenvectors. $\sigma = 3$. KN (UK) stands for When attributability is measured with (without) knowledge of attack. Standard deviation is in parenthesis.

| Model | Key length | UK | KN | FID↓ | PSNR ↑ | IS ↑ |
|---|---|---|---|---|---|---|
| BL | N/A | 0.83 | 0.88 | 100.15 | 22.49 (1.57) | 2.90 |
| PC[32:40] | 8 | 0.65 | **0.89** | **12.35** | 21.11 (2.07) | 4.75 |
| PC[32:48] | 16 | 0.45 | 0.85 | 13.25 | 18.56 (1.91) | 4.86 |
| PC[32:64] | 32 | 0.26 | 0.79 | 27.50 | 10.70 (0.56) | 4.50 |

## 5 CONCLUSION

This paper investigated deep watermark as a solution to enable the attribution of generative models. Our solution achieved better tradeoff between attribution accuracy and generation quality than the previous SOTA that uses shallow watermarks, and also has extremely low computational cost compared to SOTA methods that require encoder-decoder training with high data complexity, rendering our method more scalable to attributing large models with high-dimensional latent spaces. **Limitations and future directions:** (1) There is currently a lack of certification on the attribution accuracy due to the nonlinear nature of both the watermarking and the watermark estimation processes. Formally, by considering both the generation and estimation processes as discrete-time dynamics, such certification would require forward reachability analysis of watermarked contents and backward reachability analysis of the watermark, e.g., convex approximation of the support of $p_{x,i}$ and $\hat{\phi}$. It is worth investigating whether existing neural net certification methods can be applied. (2) Our method extracts watermarks from the training data. Even with feature decomposition, the amount of features that can be used as watermarks is limited. Thus the accuracy-quality tradeoff is governed by the data. It would be interesting to see if auxiliary datasets can help to learn novel and perceptually insignificant watermarks that are robust against postprocesses, e.g., background patterns.

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

# A PROOF OF PROPOSITIONS

## A.1 PROPOSITION 1

Define $J_w = \nabla g(w)$, $H_w = J_w^T J_w$, and $\bar{H}_\phi = \mathbb{E}_{\alpha \sim p_\alpha}[H_{U\alpha + \sigma V\phi}]$, where $p_\alpha$ is induced by $p_w$. Let $x_\phi(\alpha)$ be a content parameterized by $(\alpha, \phi)$. Denote by $\epsilon_a = \hat{a} - a$ the estimation error from the ground truth parameter $a$. Assume that the estimate $\hat{\alpha}(\alpha)$ is computed independent from $\phi$, and $\epsilon_\alpha$ is constant. Proposition 1 states:

**Proposition 1.** $\exists c > 0$ *such that if* $\sigma \leq c$ *and* $||\epsilon_\alpha||_2 \leq c$*, the watermark estimation problem*

$$\min_{\hat{\phi}} \quad \mathbb{E}_{\alpha \sim p_\alpha}\left[||g(U\hat{\alpha}(\alpha) + \sigma V\hat{\phi}) - x_\phi(\alpha)||_2^2\right] \tag{3}$$

*has an estimation error*

$$\epsilon_\phi = -(\sigma^2 V^T \bar{H}_\phi V)^{-1} V^T \bar{H}_\phi U \epsilon_\alpha.$$

*Proof.* Let $\hat{x} := g(U\hat{\alpha}_\phi(\alpha) + \sigma V\hat{\phi})$, we have

$$\hat{x} = g(U\hat{\alpha} + \sigma V\hat{\phi}) = g(U(\alpha + \epsilon_\alpha) + \sigma V(\phi + \epsilon_\phi)).$$

With Taylor's expansion, we have

$$\hat{x} = g(U\alpha + \sigma V\phi) + J_w(U\epsilon_\alpha + \sigma V\epsilon_\phi) + o(U\epsilon_\alpha + \sigma V\epsilon_\phi)$$
$$= x_\phi(\alpha) + J_w(U\epsilon_\alpha + \sigma V\epsilon_\phi) + o(U\epsilon_\alpha + \sigma V\epsilon_\phi).$$

Ignoring higher-order terms and , we then have

$$||x_\phi(\alpha) - \hat{x}||_2^2 = ||J_w(U\epsilon_\alpha + \sigma V\epsilon_\phi) + o(U\epsilon_\alpha + \sigma V\epsilon_\phi)||_2^2$$
$$= ||J_w(U\epsilon_\alpha + \sigma V\epsilon_\phi)||_2^2 + o(U\epsilon_\alpha + \sigma V\epsilon_\phi)^T J_w(U\epsilon_\alpha + \sigma V\epsilon_\phi).$$

For any $\tau > 0$, there exists $c$, such that if $\sigma \leq c$ and $||\epsilon_\alpha||_2 \leq c$,

$$||x_\phi(\alpha) - \hat{x}||_2^2 \leq ||J_w(U\epsilon_\alpha + \sigma V\epsilon_\phi)||_2^2 + \tau$$
$$= \sigma^2 \epsilon_\phi^T V^T H_w V \epsilon_\phi + 2\epsilon_\phi^T V^T H_w U \epsilon_\alpha + \epsilon_\alpha^T U^T H_w U \epsilon_\alpha + \tau.$$

Removing terms independent from $\epsilon_\phi$ to reformulate equation 3 as

$$\min_{\epsilon_\phi} \quad \sigma^2 \epsilon_\phi^T V^T \bar{H}_\phi V \epsilon_\phi + 2\epsilon_\phi^T V^T \bar{H}_\phi U \epsilon_\alpha,$$

the solution of which is

$$\epsilon_\phi = -(\sigma^2 V^T \bar{H}_\phi V)^{-1} V^T \bar{H}_\phi U \epsilon_\alpha.$$

$\square$

## A.2 PROPOSITION 2

Consider two distributions: The first is $w_0 = \mu + U\alpha + V\beta$ where $\mu \in \mathbb{R}^{d_w}$, $\alpha \sim \mathcal{N}(0, diag(\lambda_U))$, and $\beta \sim \mathcal{N}(0, diag(\lambda_V))$. $diag(\lambda)$ is a diagonal matrix where diagonal elements follow $\lambda$. The second distribution is $w_1 = \mu + U\alpha + \sigma V\phi$ where $\sigma > 0$ and $\phi \in \{0, 1\}^{d_\phi}$. Let $g : \mathbb{R}^{d_w} \to \mathbb{R}^{d_x} \in C^1$. Let the mean and covariance matrix of $w_i$ be $\mu_i$ and $\Sigma_i$. Denote by $\bar{H}_U = \mathbb{E}_\alpha[J_{\mu+U\alpha}^T J_{\mu+U\alpha}]$ the mean Gram matrix, and let $\gamma_{U,max}$ be the largest eigenvalue of $\bar{H}_U$. Proposition 2 states:

**Proposition 2.** *For any* $\tau > 0$ *and* $\eta \in (0, 1)$*, there exists* $c(\tau, \eta) > 0$ *and* $\nu > 0$*, such that if* $\sigma \leq c(\tau, \eta)$ *and* $\lambda_{V,i} \leq c(\tau, \eta)$ *for all* $i = 1, ..., d_\phi$*,* $||\mu_0 - \mu_1||_2^2 \leq \sigma^2 \gamma_{U,max} d_\phi + \tau$ *and* $|tr(\Sigma_0 - \Sigma_1)| \leq \lambda_{V,max}\gamma_{U,max}d_\phi + 2\nu\sigma\sqrt{d_\phi} + \tau$ *with probability at least* $1 - \eta$*.*

*Proof.* We start with $||\mu_0 - \mu_1||_2^2$. From Taylor's expansion and using the independence between $\alpha$ and $\beta$, we have

$$\begin{aligned}
\mu_0 :=&\mathbb{E}_{\alpha,\beta}\left[g(\mu + U\alpha + V\beta)\right]\\
=&\mathbb{E}_\alpha\left[g(\mu + U\alpha)\right] + \mathbb{E}_{\alpha,\beta}\left[J_{\mu+U\alpha}V\beta + o(J_{\mu+U\alpha}V\beta)\right] + \\
=&\mathbb{E}_\alpha\left[g(\mu + U\alpha)\right] + \mathbb{E}_{\alpha,\beta}\left[o(J_{\mu+U\alpha}V\beta)\right],\\
\mu_1 :=&\mathbb{E}_\alpha\left[g(\mu + U\alpha + \sigma V\phi)\right]\\
=&\mathbb{E}_\alpha\left[g(\mu + U\alpha) + o(\sigma J_{\mu+U\alpha}V\phi)\right] + \sigma\mathbb{E}_\alpha\left[J_{\mu+U\alpha}V\phi\right]\\
=&\mu_0 + \sigma\mathbb{E}_\alpha\left[J_{\mu+U\alpha}V\phi\right] + \mathbb{E}_{\alpha,\beta}\left[o(J_{\mu+U\alpha}V(\sigma\phi - \beta))\right].
\end{aligned} \tag{4}$$

Let $v = V\phi$. With orthonormal $V$ and binary-coded $\phi$, we have

$$\|v\|_2^2 = \phi^T V^T V \phi = \|\phi\|_2^2 \leq d_\phi. \tag{5}$$

For the residual term $\|\mathbb{E}_{\alpha,\beta} [o(J_{\mu+U\alpha} V (\sigma\phi - \beta))]\|_2^2$ and any $\tau > 0$ and $\eta \in (0,1)$, there exists $c(\tau, \eta) > 0$, such that if $\sigma \leq c(\tau, \eta)$ and $\lambda_{V,i} \leq c(\tau, \eta)$ for all $i = 1, ..., d_\phi$, we have

$$\Pr\left(\|\mathbb{E}_{\alpha,\beta} [o(J_{\mu+U\alpha} V (\sigma\phi - \beta))]\|_2^2 \leq \tau\right) \geq 1 - \eta. \tag{6}$$

Lastly, we have

$$\|\mathbb{E}_\alpha[J_{\mu+U\alpha} v]\|_2^2 \leq \mathbb{E}_\alpha[v^T J_{\mu+U\alpha}^T J_{\mu+U\alpha} v]$$
$$= v^T \bar{H}_U v \leq \gamma_{U,max} \|v\|_2^2 \leq \gamma_{U,max} d_\phi. \tag{7}$$

Then combining equation 11, equation 5, equation 6, equation 7, we have with probability at least $1 - \eta$

$$\|\mu_0 - \mu_1\|_2^2 \leq \sigma^2 \gamma_{U,max} d_\phi + \tau. \tag{8}$$

For covariances, let $\Sigma_U = Cov(g(\mu + U\alpha))$. We have

$$\Sigma_0 := \mathbb{E}_{\alpha,\beta} \left[ (g(\mu + U\alpha + V\beta) - \mu_0)(g(\mu + U\alpha + V\beta) - \mu_0)^T \right]$$
$$= \Sigma_U + \mathbb{E}_\alpha \left[ J_{\mu+U\alpha} V diag(\lambda_V) V^T J_{\mu+U\alpha}^T \right] + \mathbb{E}_{\alpha,\beta} \left[ o(J_{\mu+U\alpha} V\beta)(g(\mu + U\alpha + V\beta) - \mu_0)^T \right]$$
$$\Sigma_1 := \mathbb{E}_\alpha \left[ (g(\mu + U\alpha + \sigma V\phi) - \mu_1)(g(\mu + U\alpha + \sigma V\phi) - \mu_1)^T \right]$$
$$= \Sigma_U + \sigma^2 Cov(J_{\mu+U\alpha} V\phi^*) + 2\sigma Cov(g(\mu + U\alpha), J_{\mu+U\alpha} V\phi + o(J_{\mu+U\alpha} V\phi)). \tag{9}$$

For $tr(\Sigma_0)$, using the same treatment for the residual, we have for any $\tau > 0$ and $\eta \in (0, 1)$, there exists $c(\tau, \eta) > 0$, such that if $\lambda_{V,i} \leq c(\tau, \eta)$ for all $i = 1, ..., d_\phi$, the following upper bound applies with at least probability $1 - \eta$:

$$tr(\Sigma_0) \leq tr(\Sigma_U) + tr(\mathbb{E}_\alpha \left[ J_{\mu+U\alpha} V diag(\lambda_V) V^T J_{\mu+U\alpha}^T \right]) + \tau$$
$$\leq tr(\Sigma_U) + \lambda_{V,max} tr(\mathbb{E}_\alpha \left[ J_{\mu+U\alpha} V V^T J_{\mu+U\alpha}^T \right]) + \tau$$
$$= tr(\Sigma_U) + \lambda_{V,max} tr(\mathbb{E}_\alpha \left[ V^T J_{\mu+U\alpha}^T J_{\mu+U\alpha} V \right]) + \tau \tag{10}$$
$$\leq tr(\Sigma_U) + \lambda_{V,max} \gamma_{U,max} tr(V^T V) + \tau$$
$$\leq tr(\Sigma_U) + \lambda_{V,max} \gamma_{U,max} d_\phi + \tau.$$

For the lower bound, we have $tr(\Sigma_0) \geq tr(\Sigma_U)$.

For $tr(\Sigma_1)$, we first denote by $J_i^T$ the $i$th row of $J_{\mu+U\alpha}$, $\Sigma_{J_i}$ its covariance matrix, and $\sigma_i^2$ the maximum eigenvalue of $\Sigma_{J_i}$. Then with binary-coded $\phi$, we have

$$Var(J_i^T V\phi) = \phi^T V^T Cov(J_i) V\phi \leq \sigma_i^2 d_\phi. \tag{11}$$

Then let $g_i$ (resp. $v_i$) be the $i$th element of $g(\mu + U\alpha)$ (resp. $J_{\mu+U\alpha} V\phi$), and $\sigma_{U,i}^2$ be the $i$th diagonal element of $\Sigma_U$. Using equation 11, we have the following bound on the trace of the covariance between $g(\mu + U\alpha)$ and $J_{\mu+U\alpha} V\phi$:

$$|tr(Cov(g(\mu + U\alpha), J_{\mu+U\alpha} V\phi))| = \left| \sum_{i=1}^{d_x} Cov(g_i, v_i) \right| \leq \sum_{i=1}^{d_x} \sigma_{U,i} \sigma_i \sqrt{d_\phi}. \tag{12}$$

Lastly, by ignoring $\sigma^2$ terms and borrowing the same $\tau$, $\eta$, and $c(\tau, \eta)$, we have with probability at least $1 - \eta$:

$$tr(\Sigma_1) \leq tr(\Sigma_U) + 2\sigma tr(Cov(g(\mu + U\alpha), J_{\mu+U\alpha} V\phi)) + \tau$$
$$\leq tr(\Sigma_U) + 2\sigma \sum_{i=1}^{d_x} \sigma_{U,i} \sigma_i \sqrt{d_\phi} + \tau, \tag{13}$$

and

$$tr(\Sigma_1) \geq tr(\Sigma_U) - 2\sigma \sum_{i=1}^{d_x} \sigma_{U,i} \sigma_i \sqrt{d_\phi}. \tag{14}$$

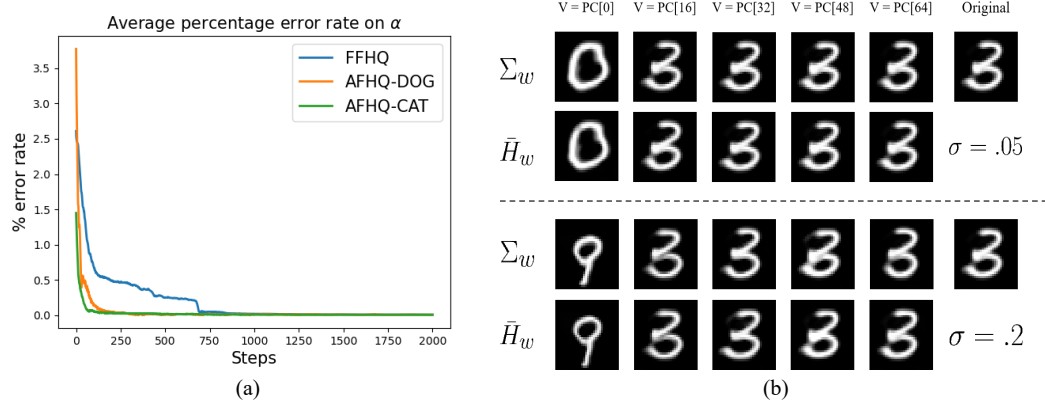

Figure 4: (a) Average percentage error rate on $\alpha$ (b) Comparison between watermarks guided by $\Sigma_w$ and $\bar{H}_w$. The editing strength for top two rows and bottom two rows are 0.05 and 0.2 respectively.

Therefore, with probability at least $1 - \eta$

$$tr(\Sigma_0) - tr(\Sigma_1) \leq \lambda_{V,max}\gamma_{U,max}d_\phi + 2\sigma \sum_{i=1}^{d_x} \sigma_{U,i}\sigma_i \sqrt{d_\phi} + \tau, \tag{15}$$

and

$$tr(\Sigma_0) - tr(\Sigma_1) \geq -2\sigma \sum_{i=1}^{d_x} \sigma_{U,i}\sigma_i \sqrt{d_\phi} - \tau. \tag{16}$$

$\square$

### A.3 PROPOSITION 3

**Proposition 3.** *For any $\tau > 0$, $\exists c(\tau) > 0$ such that if $\sigma \leq c(\tau)$,*

$$\mathbb{E}_{\alpha \sim p_\alpha} \left[ \|g(\mu + U\alpha) - g(\mu + U\alpha + \sigma V\phi)\|_2^2 \right] \leq \sigma^2 \gamma_{U,max}d_\phi + \tau. \tag{17}$$

*Proof.* We can reuse the same techniques as in the previous proofs. For some $\tau > 0$, there exists $c(\tau) > 0$, so that when $\sigma < c(\tau)$, the bound on MSE can be derived as follows:

$$
\begin{aligned}
\mathbb{E}_{\alpha \sim p_\alpha} \left[ \|g(\mu + U\alpha) - g(\mu + U\alpha + \sigma V\phi)\|_2^2 \right] &= \mathbb{E}_{\alpha \sim p_\alpha} \left[ \|\sigma J_{\mu+U\alpha}V\phi + o(\sigma(\mu + U\alpha)^T J_{\mu+U\alpha}V\phi)\|_2^2 \right] \\
&\leq \mathbb{E}_{\alpha \sim p_\alpha} \left[ \|\sigma J_{\mu+U\alpha}V\phi\|_2^2 \right] + \tau \\
&= \sigma^2 \mathbb{E}_{\alpha \sim p_\alpha} \left[ \phi^T V^T J_{\mu+U\alpha}^T J_{\mu+U\alpha}V\phi \right] + \tau \\
&\leq \sigma^2 \gamma_{U,max}d_\phi + \tau.
\end{aligned}
\tag{18}
$$

$\square$

## B CONVERGENCE ON $\alpha$

In the proofs, we assume that $\|\epsilon_\alpha\|_2$ is small and constant. Here we show empirical estimation results on SG2 and on FFHQ, AFHQ-DOG, AFHQ-CAT datasets. The results in Fig. 4(a) are averaged over 100 random $\alpha$ and 100 random $\phi$, and uses parallel search on $\alpha$ during the estimation.

## C QUALITATIVE SIMILARITY BETWEEN $\bar{H}_w$ AND $\Sigma_w$

Since computing $\bar{H}_w$ for large models is intractable, here we train a SG2 on MNIST to estimate $\bar{H}_w$. Fig. 4(b) summarizes perturbed images from a randomly chosen reference along principal

components of $\bar{H}_w$ and $\Sigma_w$. Note that both have quickly diminishing eigenvalues. Therefore most components other than the few major ones lead to imperceptible changes in the image space.

## D  ABLATION STUDY

In this section, we estimated attribution accuracy based on various attack parameters with multiple editing directions (see Tab.6,7,8,9). The image quality evaluation is available in Tab.10 and more visualizations can be found in Fig.5.

Table 6: Attributability Table of `Blurring` attack. $\sigma$ refers standard deviation of Gaussian Blur filter size 25. When attributability is measured with (without) knowledge of attack, we put results under KN (UK).

| Metric | Model | $\sigma$=0.5 | | $\sigma$=1.0 | | $\sigma$=1.5 | | $\sigma$=2.0 | |
|--------|-------|------|------|------|------|------|------|------|------|
| - | - | UK | KN | UK | KN | UK | KN | UK | KN |
| | BL | 0.88 | 0.89 | 0.87 | 0.89 | 0.87 | 0.88 | 0.85 | 0.88 |
| | PC[32:40] | 0.99 | 0.99 | 0.95 | 0.99 | 0.90 | 0.99 | 0.53 | 0.92 |
| Att. ↑ | PC[32:48] | 0.99 | 0.99 | 0.97 | 0.99 | 0.72 | 0.92 | 0.38 | 0.88 |
| | PC[32:64] | 0.99 | 0.99 | 0.73 | 0.94 | 0.51 | 0.90 | 0.32 | 0.83 |

Table 7: Attributability Table of `Noise` attack. $\sigma$ refers standard deviation of Gaussian normal distribution. When attributability is measured with (without) knowledge of attack, we put results under KN (UK).

| Metric | Model | $\sigma$=0.025 | | $\sigma$=0.05 | | $\sigma$=0.075 | | $\sigma$=0.1 | |
|--------|-------|------|------|------|------|------|------|------|------|
| - | - | UK | KN | UK | KN | UK | KN | UK | KN |
| | BL | 0.87 | 0.88 | 0.86 | 0.88 | 0.86 | 0.87 | 0.85 | 0.87 |
| Att. ↑ | PC[32:40] | 0.99 | 0.99 | 0.99 | 0.99 | 0.99 | 0.99 | 0.99 | 0.99 |
| | PC[32:48] | 0.99 | 0.99 | 0.97 | 0.99 | 0.94 | 0.99 | 0.95 | 0.99 |
| | PC[32:64] | 0.98 | 0.99 | 0.95 | 0.99 | 0.92 | 0.98 | 0.93 | 0.98 |

Table 8: Attributability Table of JPEG attack. Q refers quality metric of JEPG compression. When attributability is measured with (without) knowledge of attack, we put results under KN (UK).

| Metric | Model | Q=80 | | Q=70 | | Q=60 | | Q=50 | |
|--------|-------|------|------|------|------|------|------|------|------|
| - | - | UK | KN | UK | KN | UK | KN | UK | KN |
| | BL | 0.88 | 0.89 | 0.87 | 0.89 | 0.87 | 0.89 | 0.87 | 0.89 |
| Att. ↑ | PC[32:40] | 0.99 | 0.99 | 0.99 | 0.99 | 0.99 | 0.99 | 0.99 | 0.99 |
| | PC[32:48] | 0.99 | 0.99 | 0.99 | 0.99 | 0.99 | 0.99 | 0.99 | 0.99 |
| | PC[32:64] | 0.99 | 0.99 | 0.99 | 0.99 | 0.99 | 0.99 | 0.98 | 0.99 |

## E  SIMPLE WATERMARKING STRATEGY OF GENERATIVE MODELS

In the main paper, we studied deep watermarking methodology using SG2 and LDM. Both models have multi-layer network that maps Gaussian distribution to disentangled latent distribution. This well-trained latent distribution enables the generative models to create realistic images and embed watermarks without quality degradation as we discussed. Therefore, introducing disentanglement mapping network has become the main-stream to design generative models. However, we would also like to test the effectiveness for generative models without a disentanglement mapping by proposing a naive deep watermarking strategy. To test this, we employed BigGAN (Brock et al., 2018).

| Original | PC[32:40] | PC[32:48] | PC[32:64] |
|----------|-----------|-----------|-----------|

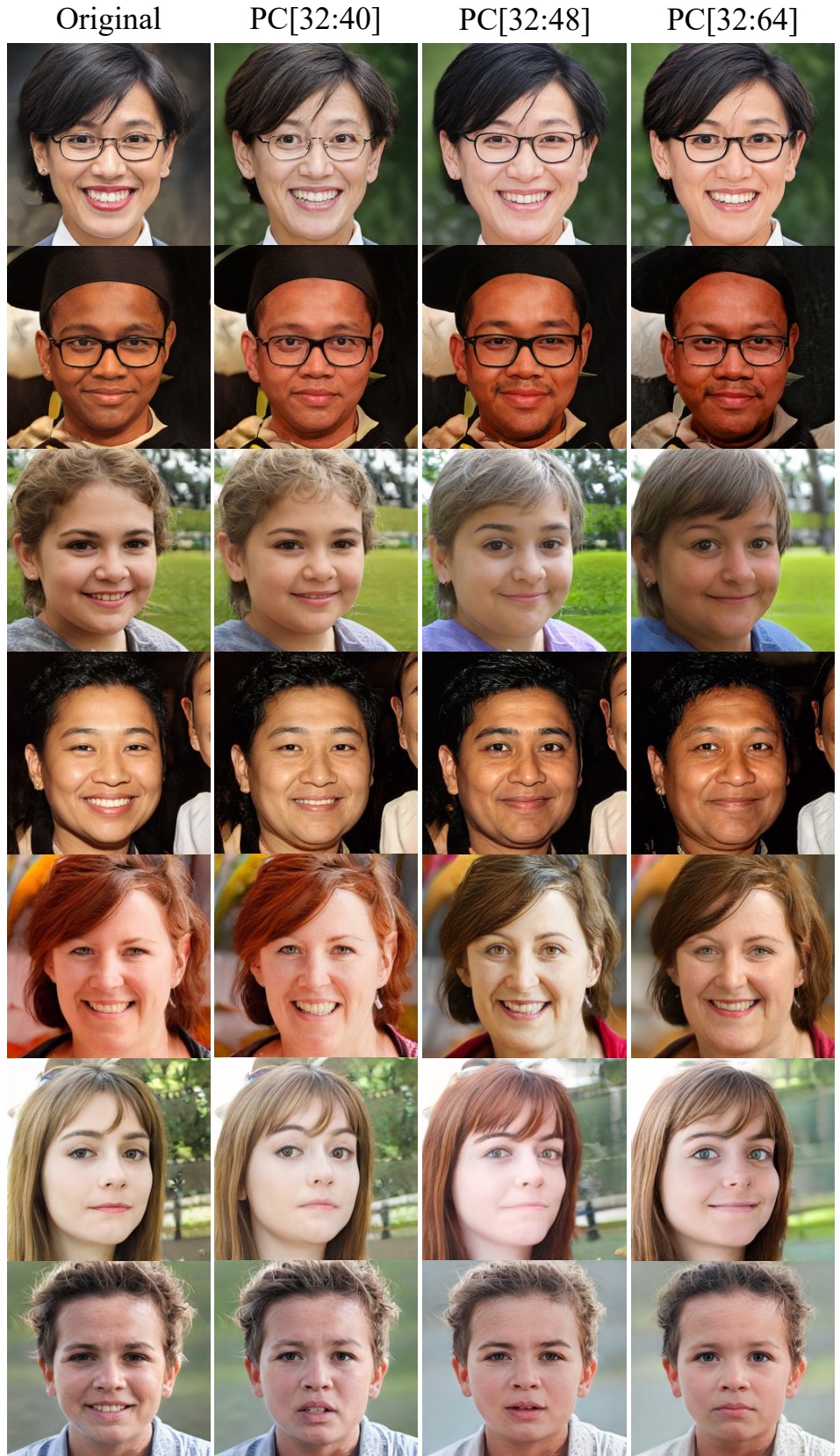

Figure 5: Qualitative Comparison of Watermarked Samples. The first column shows original images $g_0(w)$. Deep watermarked images are in the second to the last column.

Table 9: Attributability Table of combination attack. From T1 to T4, the attack parameters are composed of the weakest to the strongest attack parameters of each attack (e.g., T4 is [$\sigma_{blur} = 2.0$, $\sigma_{noise} = 0.2$, Q$_{JPEG}$=50]). When attributability is measured with (without) knowledge of attack, we put results under KN (UK).

| Metric | Model | T1 | | T2 | | T3 | | T4 | |
|---|---|---|---|---|---|---|---|---|---|
| - | - | UK | KN | UK | KN | UK | KN | UK | KN |
| | BL | 0.86 | 0.88 | 0.86 | 0.87 | 0.85 | 0.86 | 0.83 | 0.88 |
| Att. ↑ | PC[32:40] | 0.99 | 0.99 | 0.94 | 0.99 | 0.81 | 0.95 | 0.65 | 0.89 |
| | PC[32:48] | 0.99 | 0.99 | 0.74 | 0.92 | 0.52 | 0.88 | 0.45 | 0.85 |
| | PC[32:64] | 0.99 | 0.99 | 0.63 | 0.90 | 0.41 | 0.82 | 0.26 | 0.79 |

Table 10: Quality Comparison Table. Standard deviation are in parentheses. The baseline score is in the parentheses.

| Model | FID (7.24)↓ | IS (4.95)↑ | PSNR ↑ |
|---|---|---|---|
| BL$_{Blur}$ | 99.05 | 2.86 (0.35) | 21.36 (1.51) |
| BL$_{Noise}$ | 93.04 | 3.02 (0.27) | 21.57 (1.88) |
| BL$_{JPEG}$ | 97.70 | 2.91 (0.26) | 21.44 (1.64) |
| Bl$_{Combo}$ | 100.15 | 2.90 (0.23) | 22.49 (1.57) |
| PC[32:40] | **12.35** | 4.75 (0.05) | 21.11 (2.07) |
| PC[32:48] | 13.25 | **4.86 (0.08)** | 18.56 (1.81) |
| PC[32:64] | 27.50 | 4.50 (0.07) | 10.70 (0.56) |

## E.1 METHODS

Different from SG2 and LDM, BigGAN generates images $\mathcal{R}^{d_x}$ using vectors $z$ that sampled from Gaussian distribution $N^{d_z}$. Instead of simply picking random dimensions and perturbing those elements of $z$, we applied the deep watermarking method 3.Since $z \in \mathcal{N}^{d_z}$ does not have disentangled properties, this method semantically changes generated image. However, this approach also achieves high performance of attribution accuracy and FID scores.

## E.2 EXPERIMENTS

The BigGAN's input $z$ is in $N^{128}$ and $d_\phi = 32$ for the following experiments.We tested this approach on randomly selected Imagenet Deng et al. (2009) classes including cheeseburger, fountain, golden retriever, husky, Persian cat, and white wolf. For each of the classes, we generated 1,000 images for experiments. We measured attribution accuracy 1 and generation quality (FID). As shown in the table 11 and figure 6, this approach also keeps quantitative and qualitative performances of pretrained BigGAN.

Table 11: Quality changes and accuracy of the proposed method. FID-BL is baseline FID score of each class. ↑ (↓) indicates higher (lower) is desired.

| Imagenet Class | FID-BL | FID ↓ | Accuracy ↑ |
|---|---|---|---|
| White Wolf | 28.14 | 26.79 | 0.99 |
| Persian Cat | 67.25 | 65.78 | 0.98 |
| Golden Retriever | 42.16 | 40.92 | 0.94 |
| Husky | 55.55 | 49.78 | 0.97 |
| Fountain | 70.30 | 65.27 | 0.99 |
| Cheese Burger | 44.07 | 42.38 | 0.99 |

Original    Watermarked         Original    Watermarked

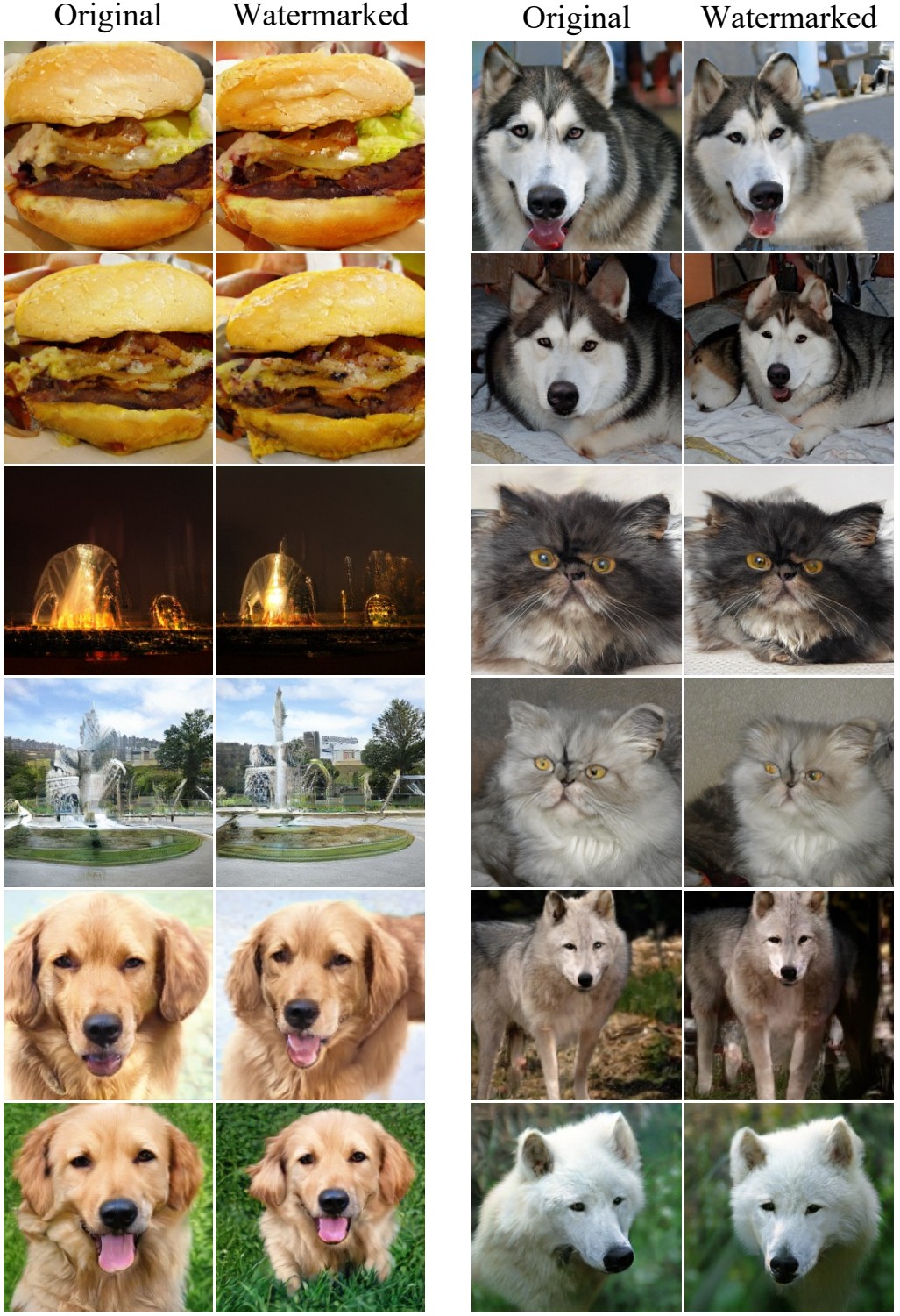

Figure 6: Watermarking Results of BigGAN: Watermarking semantically changed original images (Left) to watermarked images (Right).

