# OpenReview forum: "Deep Watermarks for Attributing Generative Models"
_ICLR.cc/2023/Conference — Submitted to ICLR 2023_

### Official Review · Reviewer_X6TA · 2022-10-25

**Confidence:** 3
**Correctness:** 4
**Technical Novelty And Significance:** 3
**Empirical Novelty And Significance:** 3
**Recommendation:** 6

**Clarity, Quality, Novelty And Reproducibility:**

The paper is clear and show goos quality.
It present a novel method.
As the code is not available, it is difficult to reproduce.


**Strength And Weaknesses:**


Strengths:

1) the paper is well written and well organised, presenting convincing evidence that it outperforms the SOTA
2) the theoretical derivations are thorough and seem also convincing

Weaknesses:
1) this authors only compare their method with the method of Kim et al. (2020). The method should be compared against more methods that explore different properties of watermarking. For instance:

Matthew Tancik, Ben Mildenhall, and Ren Ng. Stegastamp: Invisible hyperlinks in physical photographs.
In Proceedings of the IEEE/CVF Conference on Computer Vision and Pattern Recognition
(CVPR), pp. 2117–2126, 2020. [exploring printer-proof watermarking]

Shadmand, Farhad, Iurii Medvedev, and Nuno Gonc̗alves. "CodeFace: A Deep Learning Printer-Proof Steganography for Face Portraits." IEEE Access 9 (2021): 167282-167291. [exploring printer-proof watermarking]

Ruowei Wang, Chenguo Lin, Qijun Zhao, and Feiyu Zhu. Watermark faker: towards forgery of
digital image watermarking. In 2021 IEEE International Conference on Multimedia and Expo
(ICME), pp. 1–6. IEEE, 2021. [exploring resistance to watermark attack]


**Summary Of The Paper:**

This paper presents a new method to improve watermarks for generative models, particularly investigating the trade-off between attribution accuracy and generation quality. The proposed method explores latent semantic space as watermark, by mean of principal components and eigenvectors. The paper also explores how the design parameters of the method can be used to establish a practical and computationally inexpensive trade-off. Experiments show that the method outperforms the SOTA that uses shallow watermarking.

**Summary Of The Review:**

The paper is interesting and presents a convincing method for watermarking. However, the paper would benefit of comparing its results against more difficult scenarios (printer-camera) and alternative methods.

---

> ### Author Response · Authors · 2022-11-19
> **Response to Reviewer X6TA**
>
> We appreciate all the reviewers for their valuable reviews and constructive suggestions.
> * The paper is well-written (Reviewer DM6B, X6TA).
> * The proposed problem is important (Reviewer TDGU, DM6B).
> * Our methodology is novel and conceptually interesting (Reviewer TDGU, DM6B).
> * The theoretical derivation is thorough (Reviwer X6TA).
> * The paper showed trade-off between generation quality and attribution accuracy (Reviewer H68J), which provides interesting insights (Reviewer DM6B).
> * The empirical results are promising (Reviewer DM6B, X6TA).
>
> We now address individual concerns of Reviewer **X6TA** below.
>
> **1. [Compare with other SOTA watermarking model]**
>
> We would like to carefully argue that the traditional extrinsic watermarking methodology does not apply to out practical scenario. More specifically, we consider the practical use case of white-box model distribution. In white box model distribution, separate watermarking module that append after the generative model(e.g. StegaStamp [1]) is essentially trivial, since malicious user can simply remove such module from the network and avoid regulation. Instead, we propose an intrinsic watermarking module within the generative model itself. More specifically in the user end, the latent space $w_i$ of user $i$ will be provided as $w_i = U^T \alpha + \mu_i$, subject to $\mu_i = V^T \phi_i + \mu$, where $\mu$ is the mean of the latent distribution, $U$ and $V$ are two orthonormal and complementary subset of $w$ as described in paper, while $\phi_i$ being user specific constant key. By keeping $\mu$, $V$ and $\phi_i$ unknown during the model distribution, recovering the original latent space $w$ or impersonating user specific latent space $w_i$ becomes nontrivial to solve. Similarly with ours, [2] is essentially also proposing an intrinsic watermarking scheme, which makes a fair baseline comparison between our methodology.
>
> Another advantage of our model is the generalizability in terms of working with different datasets. All of the image watermarking algorithm ([3] and [4] mentioned by reviewer tdgU, and [1] mentioned by reviewer X6TA) is a deep learning based watermarking module, which essentially require the model distributor to at least retrain a separate watermarking encoder and decoder, if not apply massive hyperparameter tuning process to achieve acceptable result. Such deep learning based method often fails to generalize well in terms of image quality and attribution accuracy. However since our method only requires the model distributor to modify the pre-trained intermediate latent space $w$ through principal component analysis, it does not face such generalization issue, this is also showed empirically in table Tab.1 of the main paper.
>
> Based on the above advantages of our model, and the difference in practical use scenario, we carefully argue that the extrinsic or deep learning based watermarking methodology are out of scope of our current paper. However, we do added additional robustness experiment, extra image quality metrics and additional visual samples which can be found in appendix D.
>
> ---
> **References**
>
> [1] Matthew Tancik, Ben Mildenhall, and Ren Ng. Stegastamp: Invisible hyperlinks in physical pho-
> tographs. In Proceedings of the IEEE/CVF Conference on Computer Vision and Pattern Recog-
> nition, pp. 2117–2126, 2020.
>
> [2] Changhoon Kim, Yi Ren, and Yezhou Yang. Decentralized attribution of generative models. arXiv
> preprint arXiv:2010.13974, 2020.
>
> [3] Jiren Zhu, Russell Kaplan, Justin Johnson, and Li Fei-Fei. Hidden: Hiding data with deep networks.
> In Proceedings of the European Conference on Computer Vision (ECCV), pp. 657–672, 2018.
>
> [4] Xiyang Luo, Ruohan Zhan, Huiwen Chang, Feng Yang, and Peyman Milanfar. Distortion agnostic
> deep watermarking. In Proceedings of the IEEE/CVF Conference on Computer Vision and Pattern
> Recognition, pp. 13548–13557, 2020.

---

### Official Review · Reviewer_Dm6b · 2022-10-27

**Confidence:** 3
**Correctness:** 4
**Technical Novelty And Significance:** 3
**Empirical Novelty And Significance:** 3
**Recommendation:** 6

**Clarity, Quality, Novelty And Reproducibility:**

Whereas some aspects and the intuition of the approach are clear, some other aspects, particularly Section 3, are hard to follow. For example, the sentence "Let l : Rdx × Rdx → R be a distance metric between two images, (α̂, ϕ̂) the estimates." in 3.2 does not make clear, how the estimates parameters are related to the distance metrics. Readability can be improved by providing more details in the whole section and improving sentence structures.

Besides, the quality of the writing and experiments is good. However, some abbreviations, such as SG2, should be introduced.

Overall, the approach is novel and adds an interesting avenue to the problem of watermarking. I am not aware of previous research proposing a similar approach but I am also not deep into the watermarking literature.

I expect most parts of the approach could be implemented following the description in the paper. However, since no hyperparameters and seed are stated and no source code is provided, I expect an exact reproduction of the results to be impossible.

Smaller remarks:
- The figures, particularly in the appendix, should be added as pdf files to enable loss-less zooming.
- The statement at the end of section 4, "Attacks with large pixel-wise perturbations can be perceptually insignificant." should be supported by some reference.
- The URL in the reference of Kelly is not fitting the page margin.

**Details Of Ethics Concerns:**

I do not have any ethics concerns.

**Strength And Weaknesses:**

Strengths:
+ Tackles an important problem of generative models: watermarking the generative models (and their generated images) to allow attribution of contents to their source models.
+ Most parts of the paper are clearly written and easy to follow. Particularly the introduction section states a good entry into the problem.
+ The proposed solution of adding watermarks in the latent space instead of the image space offers an interesting avenue.
+ The empirical results promise that the model indeed beats previous approaches in terms of accuracy-quality tradeoff and induces less noticeable changes to the images.
+ The formal analysis of the accuracy-quality tradeoff provides interesting insights into the problem.

Weaknesses:
- Parts of section 3 are partially imprecise and difficult to read. Rewriting some parts of it and adding some intuition would make following it easier. Also, a list of symbols in the appendix might be worth considering.
- I expect the approach to only work for generative models with disentangled latent spaces. So for other types of GANs without using, e.g., a mapping network to entangle the dimensions, the approach is not applicable. Similarly, I am not sure if the approach works (in principle) with other diffusion models, e.g., imagen or DALL-E 2.
- I also miss some information on the computation needed to perform the attribution step. Since it needs to solve an optimization that involves forward passes through the generator, it might take some time. So I think approaches with watermarking in the image space need to train separate attribution networks, which is costly at training but fast at inference. And the proposed approach is fast at watermarking (does not need to train a separate attribution model) but the attribution process might be slow / computation costly. If this is true, the trade-off should at least be stated in the paper.
- Unfortunately, no source code is provided to reproduce the results

**Summary Of The Paper:**

The paper proposes a novel approach for watermarking generative models (GANs + latent diffusion models). While previous approaches focused on adding watermarks in the image space, the paper's approach adds watermarks in the latent space to induce subtle semantic changes to the generated images. The latent directions of change are selected as the principal components but with small variances to avoid strong semantic changes. By using these latent directions to watermark the images, the procedure promises to be computationally efficient since it needs to only compute the directions once. The paper further provides a theoretical analysis of the accuracy-quality tradeoff and empirically evaluates the approach on StyleGAN2 and Latent Diffusion models.

**Summary Of The Review:**

In summary, I like the proposed idea and direction of the paper, as well as the empirical evaluation, and think that it will benefit the community if the paper would get accepted. It tackles an important problem of generative models that becomes even more important with increasing model quality and recent advances in text-to-image synthesis. However, the quality of the writing should be improved, and details for reproduction should be added, in the best case together with the source code.

---

> ### Author Response · Authors · 2022-11-19
> **Response to Reviewer Dm6b**
>
> We appreciate all the reviewers for their valuable reviews and constructive suggestions.
> * The paper is well-written (Reviewer DM6B, X6TA).
> * The proposed problem is important (Reviewer TDGU, DM6B).
> * Our methodology is novel and conceptually interesting (Reviewer TDGU, DM6B).
> * The theoretical derivation is thorough (Reviwer X6TA).
> * The paper showed trade-off between generation quality and attribution accuracy (Reviewer H68J), which provides interesting insights (Reviewer DM6B).
> * The empirical results are promising (Reviewer DM6B, X6TA).
>
> We now address individual concerns of Reviewer **Dm6b** below.
>
> **1. [I expect the approach to only work for generative models with disentangled latent spaces]**
>
> In fact generative models without disentangled latent space can also be applied to our method. More specifically, we tested our method on BigGAN [1] with random editing direction in the Gaussian distribution of latent space $z$ for multiple labels (white wolf, Persian cat, golden retriever, husky, fountain, and cheese burger) randomly selected from Imagenet [2]. The quantitative and qualitative results are available in the Appendix E.
>
> Furthermore, we agree that applying our method to Text-to-Image (T2I) models such as DALL-E and Imgagen is an interesting direction to explore. But we carefully argue that applying our method to T2I models is out of scope of this paper. Since the inversion of T2I model remains an open challenge [3]. To elaborate the issue on T2I model inversion, the T2I model output are not only depending on latent space but also the text prompt. Assuming the text prompt is unknown, T2I model inversion will become challenging. However, we want to appeal that our methodology could be applied for T2I-watermarking as well, considering promising result shown on generative models without disentangled latent space and latent diffusion model.
>
> **2. [Concern over computational cost.]**
>
> The reason we choose to use optimization based GAN inversion is because, generally,
> learning-based inversion methods [4] and [5] cannot faithfully reconstruct the image content which is a result of inaccurate latent interpretation [6]. Particularly, such unfaithfulness becomes critical in our application, since failing to accurately recover all bits of user specific key will lead us to incorrect user. More details about the so called GAN inversion quality-time trade-off can be found in [6]. We do agree that this should be elaborated in our paper, and we revised the section 2 based on your valuable suggestion.
>
> **3. [Paper writing and Reproducibility.]**
>
> Thanks to the valuable feedback, we have revised our paper for increased readability. We have also published our code anonymously to ensure the reproducibility of our paper. The code can be found at supplementary material.
>
> ---
> **References**
>
> [1] Andrew Brock, Jeff Donahue, and Karen Simonyan. Large scale gan training for high fidelity natural
> image synthesis. arXiv preprint arXiv:1809.11096, 2018.
>
> [2] Jia Deng, Wei Dong, Richard Socher, Li-Jia Li, Kai Li, and Li Fei-Fei. Imagenet: A large-scale hi-
> erarchical image database. In 2009 IEEE conference on computer vision and pattern recognition,
> pp. 248–255. Ieee, 2009.
>
> [3] Amir Hertz, Ron Mokady, Jay Tenenbaum, Kfir Aberman, Yael Pritch, and Daniel Cohen-Or.
> Prompt-to-prompt image editing with cross attention control. arXiv preprint arXiv:2208.01626,
> 2022.
>
> [4] Guim Perarnau, Joost Van De Weijer, Bogdan Raducanu, and Jose M  ́Alvarez. Invertible conditional
> gans for image editing. arXiv preprint arXiv:1611.06355, 2016.
>
> [5] David Bau, Jun-Yan Zhu, Jonas Wulff, William Peebles, Hendrik Strobelt, Bolei Zhou, and Antonio
> Torralba. Inverting layers of a large generator. In ICLR Workshop, volume 2, pp. 4, 2019.
>
> [6] Weihao Xia, Yulun Zhang, Yujiu Yang, Jing-Hao Xue, Bolei Zhou, and Ming-Hsuan Yang. Gan
> inversion: A survey. IEEE Transactions on Pattern Analysis and Machine Intelligence, 2022.

---

> > ### Comment · Reviewer_Dm6b · 2022-11-21
> > **Response**
> >
> > I thank the authors for clarifying my questions and improving the writing of the paper. I will keep my initial score (but increase the correctness score to reflect the introduced improvements).

---

### Official Review · Reviewer_tdgU · 2022-10-28

**Confidence:** 3
**Correctness:** 3
**Technical Novelty And Significance:** 2
**Empirical Novelty And Significance:** 2
**Recommendation:** 3

**Clarity, Quality, Novelty And Reproducibility:**

Clarity: The paper could be structured better. It took me a while to sift through the actual method scattered throughout the proposition and experiment sections.

Quality: See weakness.

Originality: Good.

**Strength And Weaknesses:**

Strengths:
* The problem of source tracking the generated content of various generative models is very important.
* The proposed method is novel and conceptually interesting.

Weakness:

My main concern of this method is in the paper's evaluation setup and goals. In the paper, the authors only considered the following scenarios: (1) no post-process, (2) simple post-processing such as JPEG, noise blur etc.  Both (1) and (2) can be achieved easily by appending a blind image watermarking algorithm on top of the generated content. E.g. output = BlindWatermark(G(z), identifier). I believe the proposed baseline would actually outperform the suggested method on the simple distortions measured in this paper (See works such as [1] and [2]).

I do believe there to be advantages in encoding in the latent semantic space that the baselines [1] and [2] cannot solve. Namely, if the "distortion" itself is a "deep learning based" attack. E.g., some conditional generative model which distorts the image far in the pixel space, but retains the semantic meaning.  But this is not measured at all in the paper and defeats the purpose of encoding information in the latent space.

Other minor suggestions:
* More visual samples, and human rating results on the generated samples would help. FID alone isn't enough to determine visual sample quality.
* The structure of the paper could be improved. Proposition statements should be moved to the appendix and Section 3 should contain clearly the step-by-step explanation of the watermark encoding and decoding algorithms. This would help the less experienced reader quickly understand the proposed method.
E.g. Encoding: Find d_phi principle component vectors of matrix H ....
Decoding: Solve Equation (Sec 4.3) to estimate the watermark signal phi.


[1] Zhu, Jiren, et al. "Hidden: Hiding data with deep networks." Proceedings of the European conference on computer vision (ECCV). 2018.
[2] Luo, Xiyang, et al. "Distortion agnostic deep watermarking." Proceedings of the IEEE/CVF Conference on Computer Vision and Pattern Recognition. 2020.


**Summary Of The Paper:**

This paper proposes a method to produce watermarks on the content of generative models for the purpose of source tracking. The watermarks are added to the latent representations to produce outputs that are identifiable while preserving the quality of the distribution when compared to the original generative model.



**Summary Of The Review:**

Overall I think the paper proposes an interesting idea, and experiment results show some promise of the proposed approach. But given the major concerns over the evaluation, I do not think the paper is ready for ICLR.

---

> ### Author Response · Authors · 2022-11-19
> **Response to Reviewer tdgU**
>
> We appreciate all the reviewers for their valuable reviews and constructive suggestions.
> * The paper is well-written (Reviewer DM6B, X6TA).
> * The proposed problem is important (Reviewer TDGU, DM6B).
> * Our methodology is novel and conceptually interesting (Reviewer TDGU, DM6B).
> * The theoretical derivation is thorough (Reviwer X6TA).
> * The paper showed trade-off between generation quality and attribution accuracy (Reviewer H68J), which provides interesting insights (Reviewer DM6B).
> * The empirical results are promising (Reviewer DM6B, X6TA).
>
> We now address individual concerns of Reviewer **tdgU** below.
>
> **1. [Concerns about evaluation setup and goals]**
>
> First of all we would to explain the reason why we don't consider appending a blind image watermarking algorithm on top of the generated content, through describing the practical use case of our method.
>
> Our paper follows the same practical scenario described in [1] and [2], which aims to tackle the attribution problem of generative model output to regulate malicious use cases. Such potential threats delays the industrialization process of the generative model, as conservative model inventors hesitate to release their source code [2]. For example in 2020, OpenAI refused to release the source code of their GPT-2 [3] model due to concerns over potential malicious attempts [4], additionally, the source code of DALL-E [5] and DALL-E 2[6] are also not released for the same reason [7]. The goal of this paper is to propose a new method of regulating generative models with the hope that responsible generative model distribution can be done with minimal amount of work by model distributors. Thus, an underlying assumption for model distribution is that the model inventor will distribute the model in a white box manner. In white box model, a separate watermarking module will essentially be trivial since the user can simply remove such module and avoid regulation. Instead, we propose an intrinsic watermarking module within the generative model. More specifically in the user end, the latent space $w_i$ of user $i$ will be provided as $w_i = U^T \alpha + \mu_i$, subject to $\mu_i = V^T \phi_i + \mu$, where $\mu$ is the mean of the latent distribution, $U$ and $V$ are two orthonormal and complementary subset of $w$ as described in paper, while $\phi_i$ being user specific constant key. By keeping $\mu$, $V$ and $\phi_i$ unknown during the model distribution, recovering the original latent space $w$ or impersonating user specific latent space $w_i$ becomes nontrivial to solve.
>
> Another advantage of our model is the generalizability in terms of working with different datasets. All of the image watermarking algorithm ([8], [9] mentioned by reviewer tdgU, and [10] mentioned by reviewer X6TA) is a deep learning based watermarking module, which essentially require the model distributor to at least retrain a separate watermarking encoder and decoder, if not apply massive hyperparameter tuning process to achieve acceptable result. Such deep learning based method often fails to generalize well in terms of image quality and attribution accuracy. However since our method only requires the model distributor to modify the pre-trained intermediate latent space $w$ through principal component analysis, it does not face such generalization issue, this is also showed empirically in Tab. 1 of the main paper.
>
> Given the advantages and difference in practical use cases, we would like to carefully argue that it is not a fair comparison between our method and extrinsic or deep learning based watermarking scheme. However, we do agree that additional image quality metrics and visual samples are crucial. As a result, we added Inception score [11] as extra image quality metrics and added additional visualization samples as suggested. In the meantime, we also revised the paper flow for better readability, thanks to your valuable suggestions.

---

> ### Author Response · Authors · 2022-11-19
> **Continue Response to Reviewer tdgU**
>
> **References**
>
> [1] Changhoon Kim, Yi Ren, and Yezhou Yang. Decentralized attribution of generative models. arXiv
> preprint arXiv:2010.13974, 2020.
>
> [2] Ning Yu, Vladislav Skripniuk, Dingfan Chen, Larry Davis, and Mario Fritz. Responsible disclosure
> of generative models using scalable fingerprinting. arXiv preprint arXiv:2012.08726, 2020.
>
> [3] Alec Radford, Jeff Wu, Rewon Child, David Luan, Dario Amodei, and Ilya Sutskever. Language
> models are unsupervised multitask learners. 2019.
>
> [4] Greg Brockman, Mira Murati, Peter Welinder, and OpenAI. Openai api, 2020. URL https:
> //openai.com/blog/openai-api/
>
> [5] Aditya Ramesh, Mikhail Pavlov, Gabriel Goh, Scott Gray, Chelsea Voss, Alec Radford, Mark Chen,
> and Ilya Sutskever. Zero-shot text-to-image generation. In International Conference on Machine
> Learning, pp. 8821–8831. PMLR, 2021.
>
> [6] Aditya Ramesh, Prafulla Dhariwal, Alex Nichol, Casey Chu, and Mark Chen. Hierarchical text-
> conditional image generation with clip latents. arXiv preprint arXiv:2204.06125, 2022.
>
> [7] Pamela Mishkin and Lama Ahmad. Dall·e 2 preview - risks and limitations, 2022. URL https:
> //github.com/openai/dalle-2-preview/blob/main/system-card.md
>
> [8] Jiren Zhu, Russell Kaplan, Justin Johnson, and Li Fei-Fei. Hidden: Hiding data with deep networks.
> In Proceedings of the European Conference on Computer Vision (ECCV), pp. 657–672, 2018.
>
> [9] Xiyang Luo, Ruohan Zhan, Huiwen Chang, Feng Yang, and Peyman Milanfar. Distortion agnostic
> deep watermarking. In Proceedings of the IEEE/CVF Conference on Computer Vision and Pattern
> Recognition, pp. 13548–13557, 2020.
>
> [10] Matthew Tancik, Ben Mildenhall, and Ren Ng. Stegastamp: Invisible hyperlinks in physical pho-
> tographs. In Proceedings of the IEEE/CVF Conference on Computer Vision and Pattern Recog-
> nition, pp. 2117–2126, 2020.
>
> [11] Tim Salimans, Ian Goodfellow, Wojciech Zaremba, Vicki Cheung, Alec Radford, and Xi Chen.
> Improved techniques for training gans. In Advances in neural information processing systems,
> pp. 2234–2242, 2016.

---

### Official Review · Reviewer_h68J · 2022-11-06

**Confidence:** 3
**Clarity, Quality, Novelty And Reproducibility:** Mostly clear. Not fully known how rep…
**Correctness:** 3
**Technical Novelty And Significance:** 2
**Empirical Novelty And Significance:** 2
**Recommendation:** 5

**Strength And Weaknesses:**

The paper showed that there is a tradeoff between generation quality and attribution accuracy.

Though some aspects of the paper are interesting, many parts are still not clear. Practical use case is missing.


**Summary Of The Paper:**

This paper proposes a method to attribute generative models using digital watermarking. The paper explores the use of latent semantic dimensions as watermarks. Experiments are presented using StyleGAN2 and Latent diffusion models, which show that the approach is promising.The paper further showed that there is a tradeoff between generation quality and attribution accuracy.

**Summary Of The Review:**

Though some aspects of the paper are interesting, many parts are still not clear.

It is not clear how this approach will be useful in a practical scenario. Since watermarking is applicable in practical use cases, the authors should explain how the approach can be used practically. For example, are the authors suggesting to use a StyleGAN algorithm which also includes the deep watermarking method, and use this algorithm for generation?

It is not clear what the watermark actually is. Usually a watermark has some message or pattern embedded.

Only StyleGAN2 and Latent diffusion models have been attributed. To be practically useful, more GANs or generative methods need to be experimented.

In the post processing experiments, the post processing parameters are done for a fixed setting (example Guassian noise of -.1 standard deviation). It is also not clear what JPEG quality factor was used. Usually for post processing experiments, a range of values are considered and the analysis is then performed.

---

> ### Author Response · Authors · 2022-11-19
> **Response to Reviewer h68j**
>
> We appreciate all the reviewers for their valuable reviews and constructive suggestions.
> * The paper is well-written (Reviewer DM6B, X6TA).
> * The proposed problem is important (Reviewer TDGU, DM6B).
> * Our methodology is novel and conceptually interesting (Reviewer TDGU, DM6B).
> * The theoretical derivation is thorough (Reviwer X6TA).
> * The paper showed trade-off between generation quality and attribution accuracy (Reviewer H68J), which provides interesting insights (Reviewer DM6B).
> * The empirical results are promising (Reviewer DM6B, X6TA).
>
> We now address individual concerns of Reviewer **h68j** below.
>
> **1. [It is not clear how this approach will be useful in a practical scenario.]**
>
> We appreciate your comment and revised the introduction section based on your insightful suggestions. To elaborate, our paper follows the same practical scenario described in ([1] and [2]), which aims to tackle the attribution problem of generative model output as a mean of regulation. More specifically, recent development of generative models have enabled the creation of realistic content, however it also pose serious threat when used as malicious attempt, such as disinformation and malicious impersonation. Such potential threats delays the industrialization process of the generative model, as conservative model inventors hesitate to release their source code [2]. For example in 2020, OpenAI refused to release the source code of their GPT-2 [3] model due to concerns over potential malicious attempts [4], additionally, the source code of DALL-E [5] and DALL-E 2 [6] is also not released for the same reason [7]. This paper, as an improvement of [1], propose a new method of regulating generative models with the hope that responsible generative model distribution can be done with minimal amount of work by model distributors.
>
> In practice, we consider the scenario where the model distributor or regulator maintain a database of user specific keys which corresponds to each users' downloaded model. When malicious attempts has been made, the regulator will be able to identify the user that's responsible for such attempts by image attribution. Moreover, we assume the distributed model is white-box which makes a separate watermarking module trivial, as the malicious user can simply remove the watermarking module from the network. Instead, we choose to embed the watermarking module into the generative model itself in an intrinsic way. More specifically in the user end, the latent space $w_i$ of user $i$ will be provided as $w_i = U^T \alpha + \mu_i$, subject to $\mu_i = V^T \phi_i + \mu$, where $\mu$ is the mean of the latent distribution, $U$ and $V$ are two orthonormal and complementary subset of $w$ as described in paper, while $\phi_i$ being user specific constant key. By keeping $\mu$, $V$ and $\phi_i$ unknown during the model distribution, recovering the original latent space $w$ or impersonating user specific latent space $w_i$ becomes nontrivial to solve.
>
> **2. [It is not clear what the watermark actually is.]**
>
> Consider traditional shallow watermarking scheme applied to generated image have the form of $g_i(z)+\phi_i$, where $g_i(z)$ is the generator output without watermark, and $\phi_i$, in the sense of shallow watermarking scheme, is the user specific watermark in the image space. However in our paper, we proposed deep watermarking scheme, where the watermarked image is in the form of $g_i(\psi(z)+\phi_i)$, and the pre-watermarked image have the form of $g_i(\psi(z))$, where $\psi(\cdot)$ is the feature disentanglement functions that allow a smoother mapping to the content space, such as the mapping network for StyleGAN, the diffusion network for Latent diffusion model, etc. Thus the form of watermark is essentially, $g_i(\psi(z)+\phi_i) - g_i(\psi(z))$ (see Fig.1(a)). While the form of such watermark does not inherent a naturally observable pattern in the image space, $\phi_i$ do present as a binary code in the intermediate latent space $\psi(z)$, which can be further decoded by the GAN inversion process described in section 3.1.
>
> **3. [Is this method able to be used for other generative models?]**
>
> This paper investigate the StyleGAN2 and Latent Diffusion Model which is two state of the art image generative models, thus it's natural to focus on these two models. However we do agree with the reviewer to show the generalizability of our model on additional network. And to address this concern, we tested our method on BigGAN [8] for multiple labels (white wolf, Persian cat, golden retriever, husky, fountain, and cheese burger) randomly selected from Imagenet dataset [9]. The quantitative and qualitative results can be found in the Appendix E.

---

> ### Author Response · Authors · 2022-11-19
> **Continued Response to Reviewer h68j**
>
> **4. [Post-processing experiments should be done for various parameters.]**
>
> Following all reviewers insightful suggestion, we designed additional experiments based on a range of attack parameters. Additionally, we added the inception score as an additional image quality evaluation metrics as well as additional visual samples.
>
> The post-processing experiment considered a range of attack strength. Namely, Noising adds a Gaussian white noise of standard deviation sample from [0.025, 0.05, 0.75, 0.1]. Blurring uses a Gaussian kernel with size of 25 and standard deviation sampled from [0.5, 1.0, 1.5, 2.0]. And JPEG quality factor sampled from [80, 70, 60, 50]. For Combo, we use sequentially apply Blurring+Noising+JPEG as a deterministic worst-case attack. The results for other attacks and visual samples are available in appendix D.
>
> ---
> **References**
>
> [1] Changhoon Kim, Yi Ren, and Yezhou Yang. Decentralized attribution of generative models. arXiv
> preprint arXiv:2010.13974, 2020.
>
> [2] Ning Yu, Vladislav Skripniuk, Dingfan Chen, Larry Davis, and Mario Fritz. Responsible disclosure
> of generative models using scalable fingerprinting. arXiv preprint arXiv:2012.08726, 2020.
>
> [3] Alec Radford, Jeff Wu, Rewon Child, David Luan, Dario Amodei, and Ilya Sutskever. Language
> models are unsupervised multitask learners. 2019.
>
> [4] Greg Brockman, Mira Murati, Peter Welinder, and OpenAI. Openai api, 2020. URL https:
> //openai.com/blog/openai-api/
>
> [5] Aditya Ramesh, Mikhail Pavlov, Gabriel Goh, Scott Gray, Chelsea Voss, Alec Radford, Mark Chen,
> and Ilya Sutskever. Zero-shot text-to-image generation. In International Conference on Machine
> Learning, pp. 8821–8831. PMLR, 2021.
>
> [6] Aditya Ramesh, Prafulla Dhariwal, Alex Nichol, Casey Chu, and Mark Chen. Hierarchical text-
> conditional image generation with clip latents. arXiv preprint arXiv:2204.06125, 2022.
>
> [7] Pamela Mishkin and Lama Ahmad. Dall·e 2 preview - risks and limitations, 2022. URL https:
> //github.com/openai/dalle-2-preview/blob/main/system-card.md
>
> [8] Andrew Brock, Jeff Donahue, and Karen Simonyan. Large scale gan training for high fidelity natural
> image synthesis. arXiv preprint arXiv:1809.11096, 2018.
>
> [9] Jia Deng, Wei Dong, Richard Socher, Li-Jia Li, Kai Li, and Li Fei-Fei. Imagenet: A large-scale hi-
> erarchical image database. In 2009 IEEE conference on computer vision and pattern recognition,
> pp. 248–255. Ieee, 2009.

---

### Author Response · Authors · 2022-11-19
**Response to All Reviewers**

Thank you for all constructive reviews. We updated our main submission and highlighted major changes. Our source code is now available as supplementary material.

---

### Decision · Program_Chairs · 2023-01-20

**Decision:**

Reject

**Justification For Why Not Higher Score:**

The main concern on the lack of comparison with existing deep learning based watermarking methods is not well addressed. The authors' rebuttal fail to convince the reviewer it is not necessary to make that comparison. Also, multiple reviewers agree the more thorough post-process attacks should be included in the evaluation to test the limitation of the method.

**Justification For Why Not Lower Score:**

N/A

**Metareview: Summary, Strengths And Weaknesses:**

This paper investigates the use of latent semantic dimensions as watermarks for attributing generative models. It conducts theoretical analysis of the accuracy-quality tradeoff and evaluates the proposed method on both StyleGAN2 and latent diffusion models.

Strengths:
- It studies an important problem
- A novel watermarking approach
- Sound derivation of the method and theoretical analysis of the accuracy-quality tradeoff

Weaknesses:
- Writing quality can be further improved beside the clarification in the rebuttal
- Reviewers have concern on over-claims that external deep learning watermarking modules can be pruned trivially. It's not a sufficient argument for not comparing with those baselines. Additional comments from a reviewer after reading rebuttal:
Given more sophisticated methods (such as hiding the watermarking module with other non-watermarking layer), it is not obvious to me why it would be trivial to prune the watermarking network. The author has not provided more details on how to do this either. Therefore, I also disagree with the assessment that it is an "unfair" comparison with external watermarking modules. Further more, if pruning is considered (i.e., in a white-box setting), the authors should also evaluate watermarking removal strategies for their proposed method as well, which was not done in this paper.
- More choices of attacks should be included in evaluation.
Comments from a reviewer, "I think with carefully selected post-processing, perhaps even just horizontal flipping, watermark detection could be bypassed"

While the clarification and experiments from the rebuttal are useful to improve the clarity and addressed some concerns, this paper would require more revision to improve the writing quality and addition comparison with more baselines in order to meet the standard at ICLR.